

# Quantifying the Influence of Snow over Sea Ice Morphology on L-Band Microwave Satellite Observations in the Southern Ocean

Lu Zhou[1,2], Julienne Stroeve[3,4,5], Vishnu Nandan[3], Rosemary Willatt[4,6], Shiming Xu[7,8], Weixin Zhu[7], Sahra Kacimi[9], Stefanie Arndt[10], and Zifan Yang[11]

[1]Institute for Marine and Atmospheric Research, Department of Physics, Utrecht University, Utrecht, Netherlands
[2]Department of Earth Sciences, University of Gothenburg, Gothenburg, Sweden
[3]Centre for Earth Observation Science (CEOS), University of Manitoba,Canada
[4]Centre for Polar Observation Modeling (CPOM), University College London, London, United Kingdom
[5]National Snow and Ice Data Center (NSIDC), Cooperative Institute for Research in Environmental Sciences (CIRES), University of Colorado, Colorado, USA
[6]Centre for Polar Observation and Modelling, Department of Geography and Environmental Sciences, Northumbria University, UK
[7]Ministry of Education Key Laboratory for Earth System Modeling, Department of Earth System Science, Tsinghua University, Beijing, China
[8]University Corporation for Polar Research, Beijing, China
[9]Jet Propulsion Laboratory, California Institute of Technology, Pasadena, California, USA
[10]Alfred-Wegener-Institut Helmholtz-Zentrum für Polarund Meeresforschung, Bremerhaven, Germany
[11]School of Ecology and Nature Conservation, Beijing Forestry University, Beijing, China

**Correspondence:** Shiming Xu (xusm@tsinghua.edu.cn)

**Abstract.** Antarctic snow on sea ice can contain slush, refrozen snow-ice and stratified layers, complicating satellite retrieval processes for snow depth, ice thickness, and sea ice concentration. The introduction of moist and brine-wetted snow alters microwave snow emissions and modifies the energy and mass balance of sea ice. This study assesses the impact of brine-wetted snow and slush layers on L-band surface brightness temperatures (Tbs) by synergizing

a snow stratigraphy model (SNOWPACK) driven by atmospheric reanalysis data and a RAdiative transfer model Developed for Ice and Snow in the L-band (RADIS-L) v1.0. The updated RADIS-L v1.1 further introduces parameterisations for brine-wetted and slush snow layers over Antarctic sea ice. Our findings highlight the importance of including both brine-wetted snow and slush layers in order to accurately simulate L-band brightness temperatures, laying the groundwork for improved satellite retrievals of snow depth and ice thickness using satellite sensors such as the Soil Moisture and Ocean Salinity (SMOS) and Soil Moisture Active Passive (SMAP). However, biases

in modeled and observed L-band brightness temperatures persist, which we attribute to sub-grid scale ice surface variability and snow stratigraphy. Given the scarcity of comprehensive in situ snow and ice data in the Southern Ocean, ramping up observational initiatives in the region is imperative to provide not only satellite validation data sets but also improving process-level understanding that can scale up to improving the precision of satellite snow

and ice thickness retrievals.





## 1 Introduction

Snow on sea ice significantly influences the polar climate system by mediating mass and energy exchanges between the atmosphere and ocean and facilitating key biological and biogeochemical processes (Sturm and Massom, 2017). Recent record lows in the Antarctic sea ice extent, with departures from the 1981-2010 long-term average in excess
of seven standard deviations, underscores the urgency to understand the drivers of Antarctic sea ice variability. To better quantify the drivers, there is a need to improve our observational capacity of key sea ice variables including its overlying snow cover, a key variable needed to produce reliable ice thickness and volume data estimates (Laxon et al., 2013; Kaleschke et al., 2016). Yet, our knowledge regarding the characteristics of snow over Antarctic sea ice remains limited, partly due to the Southern Ocean's remote and harsh environment and the complexities of the
snowpacks found there.

In the Antarctic, the weight of accumulating snow can push the thin ice surface beneath the sea level (Nicolaus et al., 2009; Sturm and Massom, 2017). This results in flooding, that can lead to negative freeboards, slush formation (Jutras et al., 2016; Webster et al., 2018), and refrozen snow-ice(Merkouriadi et al., 2017; Zhaka et al., 2023). During winter, when the permeation of seawater into the snowpack becomes extensive - either infiltrating laterally at ice
floe edges or seeping through fractures in less consolidated ice (Maksym and Jeffries, 2000) - this flooding is often accompanied by high ocean heat flux melting from the bottom and redistribution and precipitation of snow occurring on the top (Ackley et al., 2020). Such conditions allow seawater's brine to infiltrate the snow, resulting in a snow-ice or slush refreeze layer, which constitutes roughly one-third of the total sea-ice mass in the Antarctic region (Maksym and Markus, 2008; Vancoppenolle et al., 2009). This layer will form shortly after flooding due to its "self-balancing"
mechanism (Sturm and Benson, 1997) and will reassert the balance and produce a zero freeboard. Notably, even when seawater flooding isn't present (Massom et al., 1998; Toyota et al., 2011), snowpacks may still house saline and damp layers at their base. Through capillary action, the brine in sea ice can ascend into the basal snow layer (Massom et al., 2001; Lewis et al., 2011), resulting in brine-wetted snow. This phenomenon can be observed when snow is deposited on the surface of new sea ice (Takizawa, 1985; Deming et al., 2010) or when sporadic warming events
amplify the ice's porosity, enabling upward brine movement (Tucker III et al., 1992). In addition to brine-wetted snow, other factors including precipitation variability, strong winds, and repeated melt/refreeze cycles contribute to the snow's complex stratigraphy (Sturm and Massom, 2017), manifesting as ice lenses, brine drainage channels, and variations in snow grain size and density (Ji et al., 2021; King et al., 2020b). Such complexities not only influence the snow's thermodynamic properties and surface albedo but also recalibrate the energy fluxes, subsequently altering
the sea ice's mass balance (Massonnet et al., 2019). Concurrently, these stratified layers induce shifts in the snow's dielectric characteristics, thereby affecting its microwave emissivity (Fuller et al., 2021), affecting the retrieved sea ice thickness parameters.

Microwave emission from snow are determined not only by its bulk properties, such as grain size, density, liquid water content, and salinity, but also by the intricacies of its stratigraphy and the characteristics of each layer.



Specifically, layers of wet or saline snow are particularly absorbent of microwave emissions (Picard and Fily, 2006; Geldsetzer et al., 2009). Even in dry snow, variations in grain size and density can significantly alter microwave emission (Tsang et al., 2004). For flooded snowpacks, the emergence of slush at the snow/ice interface or wet snow atop this slush layer can inhibit emissions from the ice beneath the snow's base (Comiso et al., 1997). Our primary challenge is to deepen our understanding of how these physical snow attributes influence microwave emissions

and backscatter. Such insights are pivotal for enhancing our capacity to accurately monitor sea ice concentration (Willmes et al., 2014), thickness (Willatt et al., 2010; Giles et al., 2008; Nandan et al., 2017, 2020), snow depth (Rösel et al., 2021), and melt onset timings (Arndt et al., 2016). With the evolving climate conditions in Antarctica, it's anticipated that snow melting and refreezing processes will become more prevalent, necessitating refined satellite retrieval algorithms for sea ice and snow properties (Raphael and Handcock, 2022; Wever et al., 2020).

Typically, snow over sea ice comprises numerous layers with different physical characteristics rather than a uniform slab (Massom et al., 2001; Sturm et al., 1998), e.g. new snow, hard slab, faceted snow and depth hoar and saline slush (Massom et al., 1998; Sturm and Benson, 1997). Established radiative transfer models, such as MEMLS (Tonboe et al., 2006), DMRT-ML (Schmidt and Wauer, 1999), and SMRT (Picard et al., 2018), despite their contributions, have been limited in representing the true complexity of snow stratigraphy over sea ice, mainly tailoring to single-

layer simulations adept for dry, cold conditions (Rostosky et al., 2018; Kilic et al., 2019). Addressing this gap, our study endeavors to enhance the understanding of snow stratigraphy's impact on passive microwave emission, leveraging more sophisticated radiation transfer models to simulate the effects of two snow layers – fresh snow overlaying a brine-wetted layer – on brightness temperatures (Tbs) over the Southern Ocean. This study, grounded in a meticulous analysis using the enhanced RAdiative transfer model Developed for Ice and Snow in the L-band

(RADIS-L; Zhou et al. (2017)), aims to foster a refined understanding of the snow stratigraphy's impact on passive microwave emission while paving the way for sophisticated satellite retrievals through nuanced radiation transfer models. The paper is structured as follows: Sec. 2 describes the observations and satellite datasets utilized in the study. Following this, the innovative approach adopted in the incorporation of new parameterisation representing brine-wetted and slush snow in RADIS-L v1.1, alongside the snow stratigraphy model (SNOWPACK) analysis, is

elucidated in Sec. 3. The subsequent sections, 4 and 5, contain critical examinations of the observed snow properties vis-à-vis the simulated Tbs against L-band satellite measurements, offering insights into model discrepancies and the outcomes of sensitivity studies. Finally, the conclusion (Sec. 6) rounds off the discussion with a contemplative reflection on the study's contributions and future research trajectories.



## 2   Data

### 2.1   In-situ measurements

### 2.1.1   ASPeCt ship-based measurements

Information on the concentration, thickness and snow cover characteristics of Antarctic sea ice has been collected from ship cruises as part of the Antarctic Sea Ice Processes and Climate (ASPeCt) programme (Worby et al., 1996; Worby and Ackley, 2000; Worby et al., 2008) (blue rectangles in Fig. 1) since the 1980s. In this paper, we used

ASPeCt sea ice thickness, ice type, snow depth and surface temperature observations available from the European Space Agency - Climate Change Initiative (ESA-CCI) sea-ice Essential Climate Variable (ECV) project, phase 2, (ESA-SICCI2). These data include collections from June 2002 through December 2019 (Kern, 2019). ASPeCt represents data along ship trajectories, and includes visual and manually conducted measurements. The temporal resolution is typically hourly but can vary by cruise. Depending on conditions, a single ship-based observation of the

sea ice generally represents an observation area with a semi-minor axis close to 1 km and a semi-major axis between an estimated 1 and 2.5 km.

### 2.1.2   Buoy measurements

#### – Snow and IMB buoy in the Weddell Sea

We also use data collected from ice mass balance buoys (IMBs) and snow buoys in the Weddell Sea from 2013-2014,

and 2016. IMBs are equipped with a thermistor string to measure sea ice temperatures with a vertical spacing of 0.02 m. According to Wever et al. (2021), the snow/ice interface is identified by the maximum of the first derivative in the vertical temperature profiles and diurnal variability in the profiles, and the accuracy in its location is estimated to be about 2-4 cm. The snow buoy measures snow accumulation using four ultrasonic sensors approximately 1.5 m above the snow/sea ice interface: hourly snow accumulation is determined by averaging the four ultrasonic sensors.

The snow buoy also measures meteorological conditions, e.g., air pressure and air temperature.

IMB 2016T41 and the collocated snow buoy 2016S31 provide data over multiyear ice (MYI) starting in January 2016 (trajectories in Fig. 1). Here, we use data from $30^{th}$ April 2016 and $1^{st}$ January 2017. Another two buoys (IMB WHOI-01 and WHOI-05), surveying the Weddell Sea during 2013 and 2014, and deployed as part of the Antarctic Winter Ecosystem and Climate Study (AWECS, ANT-XXIX/6) (Lemke, 2014) are used: IMB WHOI-05 (WHOI-01)

was installed on the ice station PS81/506 (PS81/517) and drifted with first-year ice (multi-year ice) floes, respectively (Arndt and Paul, 2018). These datasets (Wever et al., 2021) were accompanied by 2-m height weather station data from an automatic weather station (AWS) buoy, providing snow depth, air temperature, humidity and downwelling shortwave radiation. According to Wever et al. (2021), IMB snow depths are less reliable than from the sonic ranger on the AWS. Thus, for PS81/506 and PS81/517 ice stations and buoys, we use sea ice thickness from the IMB, and

rely on the AWS for snow depth and air temperature.





- **SIMBA-type buoy over Prydz Bay**

In this study, we analyzed snow depth and sea ice thickness on landfast ice in Prydz Bay using data from two Ice Mass Balance (IMB) buoy types: the US Cold Regions Research and Engineering Laboratory (CRREL-IMB),identified by names starting with 'ZS', and the Snow and Ice Mass Balance Arrays (SIMBA), with buoys named starting with 'DS'. 115 The CRREL-IMB buoys initially measure snow depth and ice thickness upon deployment. For continuous monitoring, they use an above ice acoustic sounder to track the snow surface distance and an underwater sonar altimeter for ice bottom distance. The SIMBA buoys utilize a temperature string to monitor changes in temperature profiles and detect heating-induced temperature differences, which assist in the determination of snow and ice thickness. The locations of these buoys and the corresponding sea ice parameters can be observed in Fig. 1 and Fig. B.1.

**2.1.3 Snow pits measurements from Polarstern cruise**

- **Snow density and salinity**

To parameterise the emissivity and permittivity of the brine and wet snow layers in the radiative transfer model we rely on snow properties measured at the ice stations during PS81 (green triangles in Fig. 1). A total of 60 snow pits over first-year ice (FYI: PS81/506) and multi-year ice (MYI: PS81/517) were sampled from 13 stations between 125 21 June and 2 August 2013; FYI sampled from 11 to 15 July 2013, MYI from 29 July to 2 August 2013. Vertical snow density profiles, with 3 cm thick vertical spacing from the snow surface to the snow/sea ice interface, were determined using a 100 ml density cutter. Vertical now salinity profiles for each layer and each station were measured with a salinometer after melting the snow samples Paul et al. (2017b).

- **Snow stratigraphy**

Winter snow properties over sea ice in the Weddell Sea were based on data collected from 127 snow pits during several Polarstern cruises, including ANT-XXII/2 in 2004, ANT-XXIII/7 in 2006, PS89 ANT-XXX/2 in 2013, and PS81 ANT-XXIX/6 in 2014-2015 (represented by pink triangles in Fig. 1). Snowpack stratigraphy was characterized following Fierz et al. (2009) and was primarily based on visual observations. Snow type and size for each snow layer were assessed using an 8X magnifying glass and a millimeter-scale grid card, allowing for identification of the 135 dominant grain size and type within each layer (Arndt and Paul, 2018). Layer hardness was also recorded.

**2.1.4 Ice station measurements from Sea Ice Physics and Ecosystems eXperiment II (SIPEX II) field campaign**

The snow pits and drill hole measurements utilized in this study were obtained from five ice stations conducted in the seasonal sea-ice zone off Wilkes Land, East Antarctica, between September 23 and November 11, 2012 (Toyota 140 et al., 2016). Snow stratigraphy, vertical profiles of snow temperature, grain size, density, and salinity were collected at each snow pit from three locations along 100m transects: 0 m, 50 m, and 100 m (Toyota et al., 2017; Heil et al.,



2018). Furthermore, measurements of snow depth, sea ice thickness, and freeboard were obtained from drill holes along 11 transect lines, each 100 m in length, at 1m intervals. Snow density and salinity were determined using a standard 3cm-high snow sampler with a volume of 100 cm$^3$. A total of five transects were selected for the slush
parameterisation case study in Sec. 4.3, incorporating snow and ice measurements.

## 2.2 Operation IceBridge airborne measurements

To collect more snow and ice observations for RADIS-L model validation, snow depth and sea ice thickness are compiled from the Operation IceBridge (OIB) airborne mission. Laser altimeter, Airborne Topographic Mapper (ATM), and ultra-wideband snow radar flown on OIB provide several flights transects of snow and ice thickness.
The footprint of each laser beam is 1 m and the snow radar has an 11 m across track and 14.5 m along-track footprint when flown at a 460 m nominal flight altitude (Kurtz et al., 2013).

Here, seven OIB flights over the Weddell Sea during October between 2010 and 2016 are used, including repeat surveys in 2011, 2014 and 2016 (purple lines in Fig. 1). N. et al. (2015) provides snow depth and ice thickness from the 2010 flights, with the laser freeboard data interpolated to 40 m resolution centred within the snow radar
footprints.

For the other four OIB flights, total freeboard is obtained from Kwok and Kacimi (2018) which follows the approach described by Kwok et al. (2012). Snow depths are obtained using the average from Wavelet (Newman et al., 2014) and Peakiness algorithm (Jutila et al., 2021) available through the open-source pySnowRadar package developed by King et al. (2020a) via: https://github.com/kingjml/pySnowRadar/tree/v1.1.1. Total freeboard is only calculated
above the sea level reference in the presence of open water or leads within 10 km (Kwok and Kacimi, 2018). Derived snow depths from the above two sources are shown in Fig. B.1. Fig. B.2 provides a summary of the in situ and OIB-derived snow depth and ice thickness observations.

## 2.3 Satellite Tb measurements

### 2.3.1 SMOS

In November 2009 ESA launched the L-band (1.4 GHz) Soil Moisture and Ocean Salinity (SMOS) satellite to monitor the Earth's water cycle. This fully polarized sensor (e.g. VV, HH and HV polarisations) measures emitted radiation at 1.4 GHz at multiple incidence angles from 0° to 65° (Kerr et al., 2010). One data product used here consists of an average of the vertical and horizontal polarised Tbs (L3B) (Kaleschke et al., 2012), gridded onto the NSIDC polar stereographic projection with a grid resolution of 12.5 km, using the whole incidence angle range of 0-40°).
The other data product, the L3 global polarised Tbs reprocessing RE07 product (Al Bitar et al., 2017), includes Tbs from (1) all incidence angles and (2) all polarisations in the ascending and descending orbits projected on global Equal-Area Scalable Earth Grid (EASE-Grid) 2.0 and can be freely downloaded from CATDS (available at ftp://ftp.ifremer.fr).





### 2.3.2  SMAP

NASA launched the third Soil Moisture Active Passive (SMAP) sensor in January 2015, also dedicated to observing global soil moisture. SMAP carries both a radar (active) and a 1.4 GHz radiometer (passive). The radiometer is a conically scanning radiometer at a fixed incidence angle of 40° with an approximate spatial resolution of 36 km × 47 km (Piepmeier et al., 2017). Here, we use Tbs from SMAP Radiometer Twice-Daily rSIR-Enhanced Version 2 (Brodzik et al., 2021) projected onto the EASE-Grid 2.0 at a resolution of 9 km. This data set contains twice-daily,

enhanced-resolution brightness temperature data through Scatterometer Image Reconstruction (rSIR) algorithm.

## 2.4  Auxiliary Data

### 2.4.1  AMSR-E/AMSR2

Sea ice concentration (SIC) is required as input for RADIS-L. The Advanced Microwave Scanning Radiometer for Earth Observing System (AMSR-E: 2002-2011) and AMSR2 (since 2012) provide daily estimates of SIC using

various algorithms. In this study we use the ASI sea ice algorithm (Spreen et al., 2008), which provides SIC at a spatial resolution 6.25 km). NSIDC also has AMSR-E and AMSR2 SIC datasets based on the Markus and Cavalieri (2000) algorithm, but at a coarser spatial resolution (12.5km). However, the NSIDC product additionally includes 5-day running mean averaged snow depths (Markus and Cavalieri, 2000) as well as the daily averaged Tbs for each frequency and polarisation. The snow algorithm depends on the gradient ratio of the vertical 18.7 GHz and 36.5 GHz

polarised Tbs, and is only reliable over seasonal ice and for dry snow conditions (Markus and Cavalieri, 1998).This snow depth product and Tbs are used for interpreting the surface variability, i.e. missing snow value due to leads and melting snow (Sec. 5.1).

### 2.4.2  ALOS PALSAR

To intuitively assess the effects of small-scale ice surface variability, SAR imagery is used for a case study in

Sec. 5.1. Between 2006 and 2011, Phased Array type L-band (1.27 GHz) Synthetic Aperture Radar (PALSAR) onboard the JAXA's Advanced Land Observation Satellite (ALOS) PALSAR data were acquired from different observation modes with adjustable polarisation, resolution, swath width and off-nadir angle. This study uses the newest ALOS PALSAR image data level 1.5 product from the Wide Area Observation Mode (Burst mode 1), or WB1 at the off-nadir angle of 27.1°. HH-polarised (e.g. horizontal receive and transmit polarisation) data are used,

providing five scans $350 \times 350$ km$^2$ ScanSAR images at 100 m spatial resolution. ALOS images were processed using ESA's Sentinel Application Platform (SNAP) Version 6.0 using the following steps: (i) deskewing, (ii) radiometric calibration, (iii) speckle filtering (Lee 7×7), and (iv) converted using a log scale into sigma naught backscatter coefficient ($\sigma^0_{HH}$ in dB) following Segal et al. (2020). Then, the HH-polarised backscatter were normalised to a reference angle of 35° (approximately the centre of the incidence angles (Mahmud et al., 2020) in the PALSAR





dataset): $\sigma^0_{HH}(35°)=\sigma^0_{HH}-\theta_d(\theta-\theta_{ref})$, where $\sigma^0_{HH}$ is the incidence angular dependent radar backscatter, $\theta_d$ depicts the incidence angle dependence, $\theta$ is the corresponding original incidence angle, and $\theta_{ref}$ is the incidence angle of the scene to 35°. $\theta_d$ is applied using mean frequency-specific incidence angle dependencies, -0.21 dB/1° for PALSAR, over the FYI region following Mahmud et al. (2018).

### 2.4.3  JRA55

Since not all atmosphere variables are available during in-situ and OIB campaigns, we use atmospheric fields (daily near-surface air temperature, relative humidity, wind speed, precipitation, and vertical wind profiles) from the Japanese 55-year Reanalysis (JRA55) (Kobayashi et al., 2015) for evaluation of weather influences on snow physical properties over the Weddell Sea. All atmospheric data is bilinearily interpolated into the same 12.5 km polar stereographic grid as SMOS.

## 3  Method and snow morphology parameterisation

We first briefly introduce the snow model SNOWPACK in Sec. 3.1. SNOWPACK allows for fine spatial resolution of snow stratigraphy development not always available from the snow pits. However, SNOWPACK is a model and could therefore bring additional uncertainties when using its snow variables in simulating the microwave emission along buoy trajectories; hence we only apply the relative brine-wetted depth from SNOWPACK to AWI snow buoys 220 studies. The depth of brine-wetted snow layers of the buoys deployed on landfast ice is directly measured through in-situ observations. In contrast, for ASPeCt and OIB, these measurements are inferred from calibrated negative ice freeboards. For detailed information on input parameters and their sources used in the RADIS-L model see Table A1. This work only considers the presence of a brine-wetted snow layer in cases of positive freeboard if it is explicitly confirmed by observational data.

Then, the brine-wetted and slush snow layer are parameterised into the RADIS-L model in Sec. 3.2.1 and 3.2.2, respectively, with bulk density and salinity observations (Sec. 4.1.2) from snow pit measurements deployed within the Southern Ocean.

### 3.1  SNOWPACK

The SNOWPACK model with the adaptation version over sea ice (Wever et al., 2020, 2021) is a 1-D and physical-230 based model which allows for several vertical layers for sea ice and snow. As introduced in Wever et al. (2020), SNOWPACK:

– calculates snow properties in each layer, including grain size, bond radius, sphericity, and dendricity, and also provides snow density and snow wetness, assuming equilibrium between temperature in each ice and snow layer and taking into account the brine melting point of ice;





– computes the liquid water flowing in porous media for the full range from saturated conditions (Darcy law) to unsaturated conditions. SNOWPACK is driven by air temperature, relative humidity, incoming shortwave radiation, incoming longwave radiation, wind speed, and precipitation forcings.

Here, we run SNOWPACK to simulate the snow stratigraphy evolution for buoy 2016S31 and ice stations PS86/506 and PS81/517 (see Wever et al. (2020, 2021) for details). The results for buoy 2016S31, PS81/506, and PS81/517

can be achieved with the SNOWPACK forcing datasets from the Supplement of Wever et al. (2020) and via: https://doi.org/10.5281/zenodo.4717809.

## 3.2 RADIS-L model v1.1

RADIS-L was originally designed for radiative transfer modeling of X- and L-band radiation as a function of soil moisture content (Burke et al., 1979) but was later modified to work over sea ice (Maaß, 2013) and applied to

retrievals of snow depth over thick ice (Maaß et al., 2013). Zhou et al. (2017) further modified the code to account for vertical salinity and temperature profiles in the sea ice instead of using bulk quantities. Another modification was made to differentiate ice salinity profiles as a function of ice types. The L-band Tbs were simulated using an updated version of RADIS-L v1.0, which incorporates radiative property calculations over sea ice cover. This includes aspects such as permittivity, reflectivity, and emissivity, following the methodologies outlined in Kaleschke et al. (2010) and

Maaß et al. (2013). Zhou et al. (2017) found good consistency in modelled Tbs with those retrieved from SMOS, including the observed incidence angle dependence between 0 and 40°. Recently, RADIS-L v1.0 was successfully combined with buoyancy equilibrium to retrieve sea ice thickness and snow depth over the Arctic. This method synergizes data from SMOS/SMAP with radar and laser altimeter observations and provides a more accurate and comprehensive assessment of Arctic ice conditions, as documented in research by (Xu et al., 2017; Zhou et al., 2018).

When snow weighs down ice floes sufficiently, snow could be flooded with seawater, resulting in four layers: dry snow, brine-wetted snow, slush (snow-ice), and sea ice. Even in the absence of snow flooding, the basal snow around Antarctic sea ice generally includes the presence of saline and wet layers from brine wicking upwards from the ice into the snow(Nandan et al., 2017, 2020). To enhance the simulation of complex snow properties surrounding the Antarctic region, RADIS-L v1.0 was upgraded to v1.1, adding the parameterisation of the brine-wetted snow and

slush layers in the following:

   – Our initial approach focuses on a simplified model featuring a three-layer system: dry snow, brine-wetted snow, and sea ice (encompassing both first and multi-year ice). In this context, brine-wetted snow (Sec. 3.2.1) refers to any wet and saline snow potentially present at any depth within the snowpack. Fig. 1.c illustrates examples of the hydrostatic equilibrium within this three-layer sea ice system.

– In our continued exploration, Sec. 3.2.2 explores the advanced phase of wet metamorphism, specifically focusing on the slush layer. This section characterizes the slush layer using in-situ observations and integrates these





characteristics into the parameterisation of dielectric properties in the radiation model. Building on this, Sec. 5.2 further develops our understanding by expanding the sea ice model to a four-layer schema.

### 3.2.1  Brine-wetted snow parameterisation

To characterise the thermal conductivity ($K_{bs}$ in W K$^{-1}$ m$^{-1}$) of this wet and salty snow layer (denoted as $h'_{bs}$), Lecomte et al. (2013) found Eq. 1 from Sturm and Benson (1997) was more suitable for the Southern Ocean:

$$K_{bs} = 0.138 - 0.00101 \cdot \rho'_s + 0.000003233 \cdot (\rho'_s)^2 \tag{1}$$

The thermal conductivities of ice ($K_i$) and snow ($K_s$) are taken from Zhou et al. (2017). Here, we set $z = 0$ at the base of sea ice, $z = h_i$ at the brine-wetted snow/ice interface, $z = h_i + h_{bs}$ at the dry snow, and brine-wetted snow

interface, and $z = h_i + h_{bs} + h_s$ at the snow surface. And the thermal conductivity is continuous through the $z = h_i$ and $z = h_i + h_{bs}$ interface following the Maaß et al. (2013):

$$K_i \gamma_i(z = h_i) = K_{bs} \gamma_{bs}(z = h_i) \tag{2}$$

$$K_{bs} \gamma_{bs}(z = h_i + h_{bs}) = K_s \gamma_s(z = h_i + h_{bs}) \tag{3}$$

Where $\gamma_i(z^*) = \frac{\partial T_i(z)}{\partial z}|_{z=z^*}$, $\gamma_{bs}(z^*) = \frac{\partial T_{bs}(z)}{\partial z}|_{z=z^*}$, and $\gamma_s(z^*) = \frac{\partial T_s(z)}{\partial z}|_{z=z^*}$. Given the assumption that the tem-

perature gradient is linear within the three types of layers, the temperature within the interfaces is determined by:

$$\begin{cases} T_{surf} = T_{s-bs} + \gamma_s h_s \\ T_{s-bs} = T_{bs-i} + \gamma_{bs} h_{bs} \\ T_{bs-i} = T_w + \gamma_i h_i \end{cases} \tag{4}$$

Where $T_{s-bs}$ and $T_{bs-i}$ are the interface temperatures between snow and brine-wetted snow, and brine-wetted snow and ice. The complex permittivity of this brine-wetted snow (only valid when the temperature is lower than -3°C)

is computed using the frequency dispersion model published in Geldsetzer et al. (2009):

$$\varepsilon'_{bs} = 1 + 2.55\rho_{ds} + 78.65\varphi_{bs} \tag{5}$$

$$\varepsilon''_{bs} = 27.92\varphi_{bs} + 2470\varphi_{bs}^2 \tag{6}$$

Where $\varepsilon'_{bs}$ and $\varepsilon''_{bs}$ are the permittivity and loss of brine-wetted snow, with brine volume fraction in the snow($\varphi_{bs}$) as given by Drinkwater and Crocker (1988) and $\rho_{ds}$ is the dry snow density component of brine-wetted snow $\varphi_{bs}$:

$$\varphi_{bs} = [\frac{\varphi_{bsi}\rho_b}{(1 - \varphi_{bsi})\rho_i + \varphi_{bsi}\rho_b}][\frac{\rho_s}{\rho_b}] \tag{7}$$

Where $\rho_s$ is the density of dry snow (constant 300 kg m$^{-3}$), and $\rho_i$ is the temperature dependent density of pure ice (Pouder, 1965). $\rho_b$ is the density of brine as a function of brine salinity (Cox and Weeks, 1975), which is also



a function of temperature (Poe et al., 1972). All densities are in g cm$^{-3}$. $\varphi_{bsi}$ is the temperature-dependent brine volume fraction in sea ice (Ulaby, 1982), which can be described as: $\varphi_{bsi} = S_s(-\frac{49.185}{T_s} + 0.532)$, where $S_s$ and $T_s$ are the salinity and temperature of the brine-wetted snow layer.

Normally, RADIS-L v1.1 requires information on the brine-wetted snow layer's depth, density, and salinity. Note that the percentage of this brine-wetted snow layer is determined based on two different approaches: (i) from SNOWPACK model runs when utilizing buoy observations; or (ii) through the identification of negative freeboard, a sign of flooding at the snow/ice interface leading to slush and snow-ice formation, as detailed by (Arndt et al., 2017). This latter method is employed for data derived from ASPeCt and OIB measurements. Other default settings are water temperature ($T_w$ = -1.8°C), and water salinity ($S_w$ = 33 g kg$^{-1}$).

### 3.2.2 Slush layer parameterisation

More realistically, when the snow slush is formed shortly after flooding, water-saturated snow conducts heat far better than dry snow, resulting (under freezing conditions) in a rapid refreezing layer, and converting it into snow-ice. Therefore, we treat the snow slush layer as wet and brine newly-formed snow-ice. According to the Mätzler et al. (2006), the complex dielectric constant of pure ice is written as:

$$\varepsilon_i^{'} = 3.1884 + 9.1 \times 10^{-4} \cdot (T - 273.15) \tag{8}$$

$$\varepsilon_i^{''} = \frac{(0.00504 + 0.0062 \cdot \theta) \cdot \exp(-22.1 \cdot \theta)}{f} + \frac{\beta_M + \Delta\beta}{f} \tag{9}$$

Where $\theta = \frac{300K}{T}$-1, $\beta_M = \frac{B_1}{T} \cdot \frac{\exp(b/T)}{(\exp(b/T)-1)^2} + B_2 \cdot f^2$, $B_1$ = 0.02K GHz$^{-1}$, b = 335 K, $B_2$ = 1.16 $\times$ 10$^{-11}$GHz$^{-3}$.

Then, the permittivity and loss of brine in ice is adopted the equations given in Stogryn and Desargant (1985):

$$K = \epsilon_\infty + \frac{\epsilon_s - \epsilon_\infty}{1 - 2i\pi f\tau} + i \cdot \frac{\sigma}{2\pi\epsilon_0 f} \tag{10}$$

Where $\epsilon_s$ and $\epsilon_\infty$ are the limiting static and high frequency values of the real part of K, $\tau$ is the relaxation time, f the electromagnetic frequency, $\sigma$ the ionic conductivity of dissolved salts, $\epsilon_0$ the permittivity of free space (= 8.85419$\times$10$^{-12}$ F m$^{-1}$), and i = -1. See Stogryn and Desargant (1985) for more details.

As the same treatment in Zhou et al. (2017), the brine volume fraction is calculated using coefficients from Cox and Weeks (1983) if ice temperature is below -2°C or coefficients will be determined by Leppäranta and Manninen (1988). At last, the effective permittivity of this snow ice layer is determined by solution of the quadratic Polder Van Santen mixing formula as in Matzler (1998) and Mätzler and Wiesmann (1999), which is the default formulation in SMRT improved Born approximation (IBA). According to Picard et al. (2018), it is symmetrical between the scatters and the background and has been shown to be slightly better for snow (Matzler, 1996; Sihvola, 1999).

As mentioned in Calonne et al. (2011), the effective thermal conductivity ($K_{eff}$) of snow-ice was chosen to relate with snow-ice density:

$$K_{eff} = 2.55 \times 10^{-6}\rho^2 - 1.23 \times 10^{-4}\rho + 0.024 \tag{11}$$





At last, the idealized four-layers sea ice and snow configuration (inclusion of the slush or snow-ice) is further
explored in Sec. 5.2. Although Fierz et al. (2009) classified slush when the snow wetness (liquid water content)
>15%, Matzler et al. (1982) showed that even 1% of snow wetness has a significant effect on microwave emissivity.
Due to the shortage of observed slush properties, we construct three scenarios for different water and air content
(Slush I: $\theta_w = 10\%$, $\theta_a = 15\%$; Slush II: $\theta_w = 30\%$, $\theta_a = 10\%$; Slush III: $\theta_w = 45\%$, $\theta_a = 5\%$) under the context of
hi = 2.5 m and hs = 0.5 m. Thus, the dielectric permittivity of the slush $\varepsilon_{sl}$ can be estimated with a three-phase
mixing model (Gusmeroli and Grosse, 2012): $\varepsilon_{sl} = \theta_w \cdot \varepsilon_w + (1 - \theta_w - \theta_a)\varepsilon_i + \theta_a \cdot \varepsilon_a$ where $\theta_w$ is the volume fraction of
water, $i$, $a$, $w$ are the dielectric properties for the three constituents of the mixture (ice "$i$", air "$a$" and water "$w$")
and $\theta_a$ is the air content of the slush. The bulk density of slush is set at 700 kg m$^{-3}$, representing snow composed
of icy layers (Massom et al., 2001). The average slush (snow-ice) conductivity ($K_{slush}$) bulk value is set at 0.574 W
m$^{-1}$K$^{-1}$ (Sturm et al., 2002).

### 3.3 Data regulation

Due to the inherent footprint size (30~50 km) of L-band microwave satellite from SMOS/SMAP, all input parameters
(e.g. from buoys, OIB and ASPeCT) for each day made within a 40 km grid cell are used to model the Tbs from
RADIS-L v1.1 and intercompared against Tbs from SMOS/SMAP satellites.

## 4 Results

### 4.1 Winter-time snow over sea ice properties

#### 4.1.1 Snow evolution over the buoys

From late summer until 1 September, 2016, the average early autumn snow depth at the 2016S31 buoy location
remained consistent at 19 cm. However, on this date, a significant snowfall event occurred, increasing the snow depth
to 30 cm. By the end of September, it grew to over 50 cm, as recorded by the buoy (deduced as total thickness minus
ice thickness in Fig. 2a) and used in SNOWPACK simulation (Fig. B.1a)). The snow stratigraphy simulation from
SNOWPACK for each buoy location is shown in Fig. B.3. Red colours correspond to locations with melt water within
the upper and middle snowpack, indicating the existence of the wetted layer. Starting from 14-Sep-2016, depth hoar
and melt layers began forming due to rain and higher air temperatures, as reported by (Wever et al., 2020). As a
consequence, the proportion of the wet and saline snow layer, shown as green lines in Fig. 2a, increased. This led to
a rise in the ice surface temperature (not depicted in the figures), attributed to the diminishing insulating effect of
the snow cover. Concurrently, sea ice thickness gradually started thinning from its initial measurement of 2.78 m,
continuing to thin throughout the early spring due to warming conditions.

The sea ice thinning process was detected as well, accompanied by heavy snowfall (over 35 cm snow depth) starting
from 11-Sep-2013 in PS81/506 (Fig. 2b). Throughout this period, brine-wetted layers were consistently present and





became the predominant layer by mid-October. In the case of PS81/517, more extreme conditions were observed. Here, in PS81/517 thin ice (0.5 m) and thick snow (0.25 m) initiated the formation of the brine-wetted snow layer. This process continued until the entire snow column became fully saturated.

A significant event was recorded on September 15, 2010, at the ZS-2010 buoy in Prydz Bay. A rapid increase in snow accumulation during this period resulted in the snow height reaching 0.85 meters, which eventually stabilized
at 0.55 meters, as depicted in Fig. 2d. This substantial addition of snow led to flooding on the ice surface, thus creating snow-ice conditions. In the following weeks, the snow cover continued to accumulate steadily, reaching a height of 0.85 m by November 2010. The most significant negative ice freeboard was measured at -0.09 m.

### 4.1.2 Snow density and salinity distribution

Fig. 3 provides the density and salinity characteristics of six distinct snow types, derived from an analysis of snow
pits at 13 ice stations during the period between June 21 and August 2, 2013. Further detailing can be observed in Fig. B.4, which presents stratigraphic data from ice stations PS81/506 and PS81/517. The uppermost layer of the snow pack at these locations was predominantly wind slab, while the lowest layer is characterized largely by formations such as snow-ice, crust, refrozen slush, depth hoar, or a layer of rounded crystals. Across the seven snow types, no statistically significant differences are observed in median density; all observed snow densities are bounded
within the 5% and 95% percentile range of 191.3 and 390.7 kg m$^{-3}$, with a central tendency defined by a mean value of 278.7 kg m$^{-3}$. However, density of the rounded crystals and snow-ice/slush can exceed 600 kg m$^{-3}$, and is denser than the column-averaged value, with an average of 396.7 kg m$^{-3}$.

While data on salinity are less frequently available than density data, the existing records highlight a notably higher salinity within rounded crystals and snow-ice/slush due to flooding. These records show a median salinity
value exceeding 14 psu, which is significantly higher than the overall average of 10.0 psu, falling within a range marked by the 5th percentile at 0.1 psu and the 95th percentile at 38.4 psu. Given the uniformity of our dataset and the lack of significant regional variations, we utilize the mean values of snow density and salinity as standard representations for the Southern Ocean's snow conditions. Consequently, to initially portray the brine-wetted snow scenarios, we select a representative density of of 396.7 kg m$^{-3}$ and salinity index of 10.0 psu for the snow-ice/slush
or rounded crystal layer, defining its permittivity and brine volume fraction accordingly. A detailed discussion about the choice of these bulk values and their effects can be found in in Sec. 5.2.

### 4.1.3 Snow stratigraphy frequencies

Fig. 3d depicts the frequency and relative heights of different snow stratigraphy layers observed during the winter over the Southern Ocean. The data indicates that the most common snow types are wind slab, faceted crystals, ice
crust, and snow-ice; these types are frequently found in the Antarctic region(Massom et al., 2001).

In contrast, decomposing and fragmented particles of snow appear less frequently during this season. The wind slab (precipitation particles), often found in the uppermost layer of the snow, mainly results from precipitation events,



consequently leading to the formation of a medium to high-density hard layer, as can be seen in Fig. 3. Beneath this layer, various types of snow such as faceted crystals, ice crust, rounded crystals, and depth hoar can be present, each

with the potential to be located at any level within the snowpack. Although it occurs infrequently (less than 8% of snow pits), a slush of seawater and snow is most commonly found near the bottom of the snow strata, comprising approximately the lower 20% of the structure. These layers form intriguingly; they develop when seawater infiltrates the crevices, creating brine drainage channels, which ultimately saturate the underlying snow. The dynamic process extends beyond saturation. Another notable contributing factor to this moisture is the capillary wicking-up process,

a detailed description of which can be found in Figure 6 of Massom et al. (2001). Thus, the resulting slush on the sea ice undergoes freezing, transforming into saline snow-ice, typically a consequence of seawater flooding. Additionally, the internal ICE crusts within the snow layer (Fig. 3) are formed by internal snow melt/refreeze processes, a common feature in the Antarctic snowpack. However, the introduction of water (whether from melting or rain-on-snow events) can add to the complexity and inhomogeneity of the snow stratigraphy, as noted by (Nandan et al., 2020).

**4.2    Impacts of brine-wetted snow on the Tbs measurement**

The study assumes that both the observations and simulations accurately represent the average snow properties for each ice floe. To acquire satellite Tb values, we use the nearest satellite grid point to the buoy locations, under the premise that the hydrostatic balance remains constant within the same floe and grid cell. Essentially, this means that the conditions observed at the buoy location are representative of the entire grid cell, ensuring that the satellite

Tb data is a valid proxy for the conditions across the whole floe. This premise is crucial for aligning and comparing satellite data with in-situ buoy measurements.

Heavy snow accumulation and the formation of a brine-wetted snow layer resulted in a notable decrease in SMOS Tbs. Specifically, at the 2016S31 buoy, Tbs dropped from 248.5K on September 14, 2016, to 220.2K by October 10, as shown in Fig. 4a. This decrease in Tbs, occurring despite stable sea ice concentration (depicted in light blue in Fig.

2a), is likely attributable to the newly formed brine-wetted snow layer. In the latter part of spring (mid-November), there is a discernible declining trend in Tbs. This trend aligns with observed changes in snow melt, ice thickness, and ice concentration dynamics. A similar pattern is observed with the PS81/506 buoy data (refer to Fig. 4b and Fig. B.3b). In early September at this location, snow accumulation exceeded 0.4 m, leading to a "flooding" scenario on the 0.8 m thin ice surface, which subsequently resulted in the formation of a brine-wetted layer. Due to the

nearly constant ice concentration approximating 100%, the Tbs reduction from 243.8 K (11-Sep-2013) to 226.1 K (21-Oct-2013) cannot be solely linked to an increase in open water. Notably, the depth of the brine-wetted layer in PS81/506 (Fig. 2b) becomes more pronounced around the onset of the austral spring. This correlates with the observed decrease in Tbs and an increase in surface temperatures.

In contrast, the changing Tbs in PS81/517 (Fig. 4c) do not show a clear trend, despite the presence and expansion

of the brine-wetted layer by late winter (as depicted in Fig. 2c). A similar situation is noted in September 2010 at the ZS-2010 buoy. Even though there are clear signs of snow-ice formation, as indicated by the negative freeboard





data in Fig. 2d, the changes in Tbs are not distinct (refer to Fig. 4d). However, this scenario begins to change in October 2010. This period marks the start of a gradual decrease in Tbs, coinciding with an increase in upper snow depth while maintaining a sea ice concentration of around 90%. Notably, a continuous decline in Tbs is observed,

driven by the increasing snow depth on the ice, which reaches a height of 0.85 m by October 10, 2010. This decline precedes a phase of reduction that begins around mid-December.

### 4.2.1   Tbs validation in buoy observations

#### – AWI snow buoy

As described in Sec. 3.2, we use RADIS-L v1.1 to simulate the Tbs within $40\times40\text{km}^2$ regions, using as input to the

model the sea ice thickness, snow depth, ice surface temperature from buoys, sea ice type, sea ice concentration, and the relative brine-wetted depth from SNOWPACK (Table A1). The Tbs comparison between the simulation and SMOS satellite is shown in Fig. 4 along the 2016S31, PS81/506 and PS81/517 trajectories. During the austral winter of early September 2016, SMOS (represented by black lines) observed a decrease in Tbs. However, the RADIS-L v1.0 model (blue lines) was unable to capture these reductions. In contrast, RADIS-L v1.1 (depicted in crimson

lines) accurately simulates the Tbs changes. Beginning on 29-Aug-2016 (Fig. 4a), the Tbs modeled by RADIS-L v1.1 diverge from those by RADIS-L v1.0, indicating the critical role of the brine-wetted layer in simulating Tbs over time. Overall, RADIS-L v1.1 shows a strong correlation with SMOS data ($r^2$ of 0.682) for this buoy. For buoy PS81/506, the accuracy of RADIS-L v1.1 is notable, with an increase in $R^2$ from 0.034 (RADIS-L v1.0) to 0.560 (RADIS-L v1.1). Notably, RADIS-L v1.1 reduces the overestimation biases seen with RADIS-L v1.0, particularly in

October 2013 (Fig. 4b), which aligns with the formation of the melt layer. Although the significant improvements are not seen for buoy PS81/517, the simulated Tbs still remain correlated with SMOS, with an $R^2$ of 0.252.

Furthermore, SMAP Tbs are also modelled for both horizontal (Fig. 5a) and vertical (Fig. 5b) polarisations at a fixed 40-degree incidence angle. The grey shading represents one standard deviation of Tbs from the SMOS RE07 product obtained from multiple incidence angles ranging from $2.5°$ to $62.5°$ at an interval of $5°$. There are notable

observational differences between SMOS and SMAP, especially for vertically polarized Tbs, where SMOS readings are approximately 2.9 K higher than SMAP. Huntemann et al. (2016) also found that SMOS yielded higher Tbs than SMAP in both polarisations (about 5 K). In comparison with SMAP, the simulated Tbs from RADIS-L v1.1 suggest a larger bias in the vertical polarisation. However, despite these positive biases and greater variability at vertical polarisation, the modelled Tbs maintain a high correlation with SMAP, with all $r^2$ values exceeding 0.64.

#### – Buoys on land-fast sea ice

To strengthen the validation of Tbs, we have extended the dataset by incorporating detailed observations from Prydz Bay, capitalizing on the enhanced capabilities of RADIS-L v1.1. This effort involves incorporating data on sea ice thickness, snow depth, and ambient air temperature recorded by an array of SIMBA-type buoys strategically





positioned throughout the bay. To further advance our research, additional data dimensions have been integrated.
These include ice type classifications provided by OSI-SAF, detailed ice concentration statistics sourced from ASI,
and estimates of the brine-wetted layer depth, which are based on negative ice freeboard measurements. These
additional data aspects are elaborated upon in Table A1.

In the following validation workflow, we conduct Tbs simulations following the trajectory of the ZS-2010 buoy. This
allows for a critical juxtaposition with SMOS measurements as illustrated in Fig. 4d. Notably, starting from mid-
September, the Tbs exhibit significant fluctuations. These are primarily attributed to increased snow accumulation
and recurring flooding events. A comparative study between the v1.1 and v1.0 models reveals marked differences
in Tbs, particularly in the assessment of brine-wetted snow. The v1.1 model demonstrates a notably better fit with
the observed data, characterized by a nearly perfect slope and an $r^2$ value approximating 0.36. To further validate
the RADIS-L v1.1 model, we undertake a comprehensive evaluation using datasets from an extensive network of
SIMBA-type buoys deployed across Prydz Bay between 2010 and 2018, including ZS and DS buoys. Our analysis,
illustrated in Fig. B.6, highlights the alignment of the v1.1 model with SMOS measurements. This is evidenced by
a strong correlation and a slope exceeding the 0.7 threshold, confirming our initial hypotheses and expectations.

### 4.2.2   Tbs validation in ship-based and airborne observations

Similar to buoy comparisons, the primary inputs for ASPeCt-based validation encompass parameters such as sea
ice thickness, snow depth, ice type, ice surface temperature, and concentration, derived from ASPeCt observations
(refer to Table A1). Given the SNOWPACK limitations in non-buoy applications, we adopt negative ice freeboard
as an indicator of brine-wetted snow depth. This time, ASPeCt's measurement range extends beyond the Weddell
Sea to include the Bellingshausen Sea and the South Indian Ocean, as marked by the blue squares in Fig. 1. The
Tbs modeled based on ASPeCt data (see Fig. 6a) align closely with those captured by SMOS. This validates the
performance of both RADIS-L v1.0 (shown in grey) and v1.1 (in black), with the $r^2$ values exceeding 0.9. While
RADIS-L v1.0 shows a marginally better correlation, RADIS-L v1.1 is notable for its lower positive bias in the
simulated Tbs.

Along the OIB tracks, Tbs were simulated using OIB sea ice thickness, snow depth, KT19 ice surface temperatures,
ASI sea ice concentration, OSI-SAF ice type, and brine-wetted depth determined from negative ice freeboard data
(as seen in Fig. 6b and 6c). The snow depth (Fig. B.1) from these seven campaigns shows large spatial and temporal
variability. For instance, on October 30, 2010, some snow depth measurements reached as high as 2 m, with an average
of 0.52 ±0.35 m. On October 20, 2014, snow depths peaked at 1 meter, averaging 0.49 ±0.16 m. Meanwhile, other
measurements varied between 20 cm and 45 cm. In terms of Tbs simulations, RADIS-L v1.0 tends to overestimate
the SMOS Tbs, with an $r^2$ about 0.31 and a mean bias of 7.4 K. In contrast, RADIS-L v1.1 significantly reduces these
overestimations, increasing the $r^2$ to 0.45 and demonstrating no statistically significant bias, although the mean of
the clusters shown in Fig. 6c is approximately 1.5 K higher than the SMOS data. Despite these improvements, closer
scrutiny of some simulations, particularly the data recorded on 28-Oct-2010, reveals discrepancies when compared





to satellite observations. This observation suggests a need for further investigation into small-scale ice and snow surface characteristics, which can be corroborated through SAR satellite imagery and reanalysis datasets.

## 4.3 Examining the effects of slush snow in SIPEX II case study

In Sec. 3.2.2, we delve into the most extreme stage of wet metamorphism after flooding: snow slush. We examine the unique impacts of slush and flooding snow within the context of brine-wetted snow layers. This analysis is supported by observations from five snow and ice transects located near Wilkes Land. In this study, key attributes of slush snow, including depth, density, and salinity, are compiled into mean bulk values based on Snow Pit assessments. These assessments provide insightful data: for instance, average slush snow depths at Ice Stations 2, 3, and 4 are found to be 3 cm, while at Stations 6 and 7, the depth averages 1 cm. Across these stations, the combined mean density and salinity are calculated to be 481 kg m$^{-3}$ and 9.83 psu, respectively.

Fig. 7 offers valuable insights, highlighting key metrics such as air temperature, ice thickness, snow depth, and freeboard, with a focus on critical statistical values (e.g. quartiles and medians values). Notably, each station recorded negative freeboards. Specifically, Station 4 reported the thinnest median ice thickness at 1.38 m, contrasted with a median snow depth of 0.48 m, resulting in a median negative freeboard of -0.02 m. In contrast, Station 7 displayed more substantial median values, with an ice thickness of 4.87 m and a snow depth of 0.51 m. Tbs simulations, as outlined in Table A1, were systematically evaluated both with and devoid of the slush snow parameterisation. When compared to SMOS-derived Tbs observations (illustrated in Fig. 7), RADIS-L v1.0 simulations consistently demonstrated an overestimation bias of 8.8 K. However, this bias was significantly reduced to 2.8 K in RADIS-L v1.1, particularly after incorporating the slush snow layer, thus achieving closer alignment with the SMOS datasets, especially at Stations 4 and 6. Furthermore, leveraging detailed snow morphology data from SIPEX II, the enhanced RADIS-L v1.1 model rectifies key discrepancies in existing radiation transfer models over the Southern Ocean. This improvement substantially increases the accuracy of ice and snow parameter retrieval in polar regions. The advancements made with this updated model open promising avenues for more refined and nuanced analyses in future research efforts.

## 5 Discussion

### 5.1 Sub-grid-scale surface variability

On 28-Oct-2010, OIB flew from the corner of the Antarctic Peninsula in the northern Weddell Sea, starting at -75.0°S/-38.1°W to -64.0°S/-42.3°W (Fig. 8a). Along this flight path, notable discrepancies were observed between RADIS-L v1.1-simulated Tbs and SMOS measurements. This was particularly evident over the eastern region (marked as the grey area in Fig. B.6g), where RADIS-L v1.1 overestimated the SMOS Tbs by an average of 15 K. One potential explanation for this bias is an incorrect sea ice concentration value, potentially influenced by atmospheric



conditions. The 89 GHz channel, used in the ASI algorithm, is known to be affected by liquid cloud water. JRA55
data for 28 October 2010 indicate high atmospheric humidity and warm air temperatures (Fig. B.7). Additionally,
AMSR-E Tbs (Fig. B.6) suggest surface melting over the eastern portion of the OIB flight path.

The overestimation of sea ice concentration is directly observed from ALOS HH-polarised PALSAR backscatter
over the eastern region of OIB track on 29 October 2010 (Fig. 8d and e). The brighter images in Fig. 8d and e
represent the larger backscatter over the eastern region of the track than the western one (30 October 2010, Fig. 8b
and c) and imply the presence of leads within eastern sea ice region. Moreover, the mean SAR backscatter under
the OIB footprints (Fig. 8f) shows different peaks for these two regions. On 30 October 2010, two backscatter modes
were recorded at -12.4 and -14.9 dB. In contrast, on 29 October 2010, the first peak occurred at -20.8 dB, primarily
from leads, while the second peak, consistent with October 30 observations, averaged -13.1 dB, indicating sea ice.

In summary, the JRA55 atmospheric reanalysis combined with Tbs from various AMSR-E bands indicates a
significant presence of moisture in the air and melting of surface snow over the eastern section of the OIB flight
path. Furthermore, HH-polarised ALOS images indicate leads within the ice pack. High resolution (6.25 km) ice
concentration AMSR-E datasets can not resolve the small-scale ice variability. This limitation leads to an overesti-
mation of sea ice concentration, which in turn causes an overestimation of the simulated Tbs. This highlights the
need for careful consideration of small-scale ice features in the analysis of satellite data and the modeling of sea ice
parameters.

## 5.2   Slush layer and flooding effects

For a more accurate representation of snow structures around Antarctica, a detailed four-layer configuration is ideal
for investigating the effects of slush on surface radiative properties. This configuration includes dry snow atop a
brine-wetted layer, followed by a slush layer, and finally, a sea ice layer at the bottom. As explained in Sec. 3.2.2,
we consider three distinct scenarios (illustrated in Fig. 9a-c: Slush I, II, and III) with varying water and air contents
within the slush layer. The slush layer depth is determined by the percentage of slush within the brine-wetted snow.
We also construct the slush layer with different brine-wetted snow depths (20, 40, 60, and 80% of the entire snow
depth). With a constant ice thickness of 2.5 m and snow depth of 0.5 m, the simulated Tb is 255.4 K in the absence
of any brine or slush layers. The inclusion of brine-wetted layers results in lower Tbs, decreasing to 253.7, 252.7,
252.1, and 249.5 K as the brine-wetted snow depth increases. The inclusion of the slush layer further reduces the
Tbs depending on the slush properties. Here, we found that higher water content (and lower air content) in the
slush results in higher simulated Tbs (Fig. 9d), making the Tbs more akin to ice than to snow. Fig. 9d also explores
the Tbs changes under different slush depths for each scenario. A larger slush depth reduces the simulated Tbs due
to a decrease in snow-ice temperature, resulting from less insulation provided by shallower dry snow. Moreover,
as the slush layer thickens, the Tbs decrease more significantly in Slush I (with the least snow wetness) and in
the deepest brine-wetted snow layer (80%) compared to other scenarios. Specifically, in Slush III (with the wettest
snow), as it increasingly occupies the brine-wetted snow layer up to 100% of the total snow depth, the simulated Tbs





converge to a consistent value of 226.5K. However, the Tbs from other wet slush layers (Slush I and II) vary when the whole brine-wetted layer is slush. For example, in Slush I, the final Tbs range from 186.2 to 192.4 K. Therefore,
more water content in the slush results in less sensitivity to the slush depth. It is clear that the increasing depth of the slush layer or the decreasing depth of the brine-wetted layer corresponds to a non-linear reduction of Tbs in Fig. 9d. Additionally, with the same proportion of the slush layer, the spread of Tbs among different depths of the brine-wetted layer becomes more pronounced in scenarios with less water content in the slush.

To conclude, simulations incorporating a more complex snow stratigraphy, including a slush layer, further reduce
the modeled L-band Tbs, highlighting the importance of accurately representing snow layers in such models.

## 5.3 Sensitivity in bulk parameters

The primary default settings used in the RADIS-L v1.1 simulations are snow density and salinity from the in-situ measurements around the Southern Ocean. The following examines the effects and sensitivity from using these bulk values and the schemes of sampling in the Tbs simulations.
Fig.10a and b denote the Tbs values for different snow densities, salinity and percentage of brine-wetted snow layers for a 2.5 thick ice floe covered by 0.5 m of snow. The snow density and salinity are chosen within the 5% and 95% range of PS81 ice station measurements. Fig.10 suggests that snow salinity and density are inversely correlated to L-band Tbs. Specifically, Tbs decrease by approximately 2.5 K with an increase in snow density from 250 to 400 kg m$^{-3}$. Similarly, Tbs reduce by more than 5 K when snow salinity increases from 2 to 10 g kg$^{-1}$. Notably,
the impact on L-band Tbs from changes in snow salinity is more pronounced than that from density. This can be attributed to the greater variation in the complex dielectric constant of brine-wetted snow due to salinity, as illustrated in Fig. B.8. Furthermore, the extent of Tb reduction is influenced by the percentage of brine-wetted depth within the snow. Generally, a higher percentage leads to a greater reduction in Tbs. However, for thinner brine-wetted layers, the relationship between snow properties and Tbs exhibits non-monotonic (convex) curves. The
possible explanation is that large insulation from the thin brine-wetted layer results in high temperature within the snow; thus, the permittivity of that layer becomes highly sensitive to temperature variations, exhibiting larger values at warmer temperatures (as shown in the notching curves between temperature and dielectric constant in Fig. B.8). This results in increased sensitivity of Tbs. The non-monotonic relationship between microwave observations and the impact of snow salinity remains an open area for discussion and warrants further investigation.
One phenomenon deserves notice: when the brine-wetted layer is thinner than 20% of the entire snow depth, Tbs can be lower than those from thicker brine-wetted layers. However, additional research is needed to fully understand the effects of brine-wetted layer characteristics on L-band Tbs. One noteworthy observation is that when the brine-wetted layer constitutes less than 20% of the total snow depth, However, Further exploration demands more detailed data, including a deeper understanding of the significant influence regional-dependent snow density and thermal
conductivity exert on sea ice growth, as referenced in (Arndt, 2022). However, such an extensive inquiry lies beyond the scope of this study.





In addition to the study regarding the sensitivity of default parameters, we also examine the sampling schemes, i.e. one or multiple sea ice samples (Fig.10c and d). Here, a set of 5000 samplings is generated under the context of (i) constant standard deviation (STD) for snow depth and ice thickness ($\text{STD}_{hi} = 0.5$ m, $\text{STD}_{hs} = 0.1$ m), and (ii) mean value-dependent (deduced from OIB measurements) standard deviation of ice and snow ($\text{STD}_{hi} = 0.63 \times$hi, $\text{STD}_{hs} = 0.36 \times$hs) through Monte-Carlo lognormal distribution perturbations. By applying these sampling schemes across various ice thickness and snow depth scenarios, we present the simulated Tbs for five sea ice and snow conditions using violin plots in Fig.10. The blue squares and the three horizontal lines (Fig.10c and d) represent the medians, means, and +/- standard deviations in simulated Tbs following the Monte-Carlo perturbations. Meanwhile, the orange stars indicate the Tbs values from the mean hi and hs (representing a single sampling condition). The constant STD perturbation (Fig. 10c) indicates that only one sample always overestimates the average of Tbs, especially over thin ice; these biases would be negligible over the thick ice. Similarly, value-dependent sampling (Fig. 10d) demonstrates the biases from significant overestimation to minor underestimation when ice thickness deepens. This type of sampling depicts a more realistic lognormal distribution of Tbs compared to the constant STD scheme. Thus, this sensitivity study suggests that more measurements or observation inputs would improve the accuracy in simulated Tbs or other passive microwave parameters. Moreover, thick ice and snow are less susceptible to issues of undersampling. Thus, the dramatic decline of sea ice thickness in the 21$^\text{st}$ century for both the Arctic and Antarctic (Mallett et al., 2021; Kacimi and Kwok, 2022) will continue to challenge the validity of microwave satellite remote sensing retrievals. This challenge is particularly pronounced if the algorithms rely predominantly on a limited number of sea ice measurements for training and calibration.

## 6   Conclusions

In this research, we examine the nuanced effects of Antarctic snow stratigraphy on the radiative attributes of ice surfaces, with a particular focus on brine-wetted and slush snow layers. By incorporating advanced parameterisations in RADIS-L v1.1, we have achieved significant advancements in accurately representing observed ice surface Tbs. This progress establishes a solid basis for more precise future research and applications in the realm of polar ice surface studies.

The substantial snow cover and complex snow-ice interaction in Antarctica often lead to basal snow having high salinity and moisture content. This frequently results in ice-surface flooding or the formation of snow-ice, not limited to the snow-ice interface, as indicated by Webster et al. (2018)). Utilizing data from AWI and SIMBA-type buoys, as well as simulations from SNOWPACK, we observed a progressive increase in the extent of the brine-wetted layer, especially as the ice becomes increasingly overburdened. Interestingly, this phenomenon of ice flooding is also evident in land-fast ice regions. Often, this flooding is either temporally coincident with, or preceded by, ice breaking and a reduction in ice concentration, highlighting the link between declines in Tbs and changes in the depth of the brine-wetted layer (less thermal insulation). Consequently, we have incorporated the thermal conductivity and permittivity





of brine-wetted snow, which differ from those of dry snow, into the RADIS-L v1.1 model. To validate our improved parameterisation, we used existing extensive sea ice measurements from the Southern Ocean, encompassing data from airborne platforms, in-situ buoys, and ship trajectories. We then compared these simulated Tbs with satellite data from both SMOS and SMAP. By integrating the reanalysis-driven prognostic model SNOWPACK with the diagnostic radiation model RADIS-L v1.1, we are able to demonstrate, for the first time, the critical role of the

brine-wetted snow layer in understanding the changes in radiative properties of ice surfaces.

In particular, we find that there is a significant amount of uncertainty in surface Tbs values due to input datasets such as sea ice concentration and ice surface temperature. Complicating factors include the sub-grid scale variability of the ice surface, such as the presence of leads within the ice, as described by Willmes and Heinemann (2015) and the turbulent flux exchanges occurring between the atmosphere and ocean, highlighted in Ólason et al. (2021).

Additionally, we determine that ice concentration without sub-grid scale information, i.e. leads, greatly overestimates Tbs, especially in regions with strong ice deformation. However, there is cause for optimism. The improved algorithms proposed by Paul and Huntemann (2021) and the use of different types and resolutions of satellite merged products, as suggested by Ludwig et al. (2019), show promise in capturing variations in sea ice and enhancing the accuracy of ice parameter simulations and retrievals. Our ongoing research aims to integrate these merged and high-resolution

datasets to refine the accuracy of snow depth retrieval from microwave satellites. Additionally, the integration of a detailed slush layer into RADIS-L v1.1 has proven crucial, significantly reducing the simulated Tbs in a manner closely linked to the properties and depth of the slush layer. This finding highlights the complexity of accurately simulating Tbs for thin brine-wetted snow layers and emphasizes the urgent need for detailed research to unravel the intricate relationships between snow salinity, density, and Tbs, particularly in the context of thicker layers.

The urgent need for detailed laboratory and field research is clear, especially in untangling the complexities of brine-wetted and slush layers, which are particularly prevalent in the Southern Ocean. In this region, data on such phenomena are still sparse. The scenarios we have discussed provide critical benchmarks for improving radiometer designs, as well as their calibration and validation processes in various settings. Recognizing the limitations posed by undersampled measurements is crucial; such limitations can significantly impact the precise retrieval of ice parameters

and the fine-tuning of algorithms used in L-band satellite imagery, most notably in regions with thinner ice. These limitations are particularly concerning in light of climate change and its impacts. Therefore, it's vital to enhance our data collection with consistent, detailed observations from ground-based sources, aerial surveys, and field research. Strengthening our data collection is essential for maintaining the accuracy and reliability of satellite observations in tracking and understanding changes in polar regions.

The accuracy and statistical parameters of ship observations, such as those from ASPeCt, vary based on factors like the algorithm used, satellite sensor, observation techniques, time of the expedition, and the ship's route. For example, Worby et al. (2008) mentioned that the ship was turned to avoid areas of higher concentration, rather than a straight line track that would have randomly sampled the area. Thus, the accuracy of SIC observations based on the ASPeCt protocol is within 10% (Weissling et al., 2009). This variance could be attributed to the subjective





nature of SIC retrievals around the ship by different observers. Additionally, Worby et al. (2008) estimated that thickness errors in ASPeCt observations can range from ±20% for level ice over 0.3 meters thick to as much as ±50% for ridged ice. Given these considerations, it is crucial to exercise caution when using ship-based measurements for validating Tbs. Further research is needed to fully understand the cumulative sensitivities within the RADIS-L model, particularly arising from input errors in snow and ice parameters.

Finally, building upon the successes of heritage missions like AMSR and SMOS/SMAP-type missions, the high-priority candidate mission Copernicus Imaging Microwave Radiometer (CIMR) (planned launch in 2025+ by ESA: http://www.cimr.eu/) aims to provide microwave imaging radiometer measurements across a broad spectrum, from 1.4 to 36.5 GHz, encompassing L, C, X, Ku, and Ka bands (Scarlat et al., 2020). Several studies (Kilic et al., 2020; Jiménez et al., 2021) have already stated the potential performance of the CIMR instrument and estimated

the retrieval precision, including different sea ice parameters. Therefore, the algorithms and findings presented in this paper have promising applications for simultaneous and consistent ice parameter retrievals using CIMR, considering the large frequency coverage, improved spatial resolution, and high radiometric precision. The CIMR mission is expected to facilitate a more comprehensive understanding and retrieval of complex snow stratigraphy and properties over sea ice at both poles. This will be invaluable for further development and validation of algorithms,

contributing significantly to our understanding of polar regions.

*Code availability.* The code developed for this study can be found at https://doi.org/10.5281/zenodo.10003441, (Zhou and Xu, 2023)

*Data availability.* Snow and sea ice dataset from ASPeCt ship-based measurements are available via: https://doi.org/10. 26050/WDCC/ESACCIPSMVSBSIOV2. Ice-physics transect data obtained during the SIPEX II voyage of the Aurora Aus-

tralis, 2012 (Toyota et al., 2017; Heil et al., 2018) is available via: https://doi.org/10.4225/15/59b0c7fd5c76f and https://doi. org/10.4225/15/5a8f94c228afb. The AWS buoy and IMB data from PS81/506 (https://doi.org/10.1594/PANGAEA.933415 and https://doi.org/10.1594/PANGAEA.933417), and those two from PS81/517 ( https://doi.org/10.1594/PANGAEA.933425 and https://doi.org/10.1594/PANGAEA.933424) used in this study are available from Wever et al. (2021). All presented SIMBA-type buoy over Prydz Bay are available in PANGAEA are available from Li et al. (2023) (https://doi.org/10.

1594/PANGAEA.950178, (Li et al., 2022a), https://doi.org/10.1594/PANGAEA.950181, (Li et al., 2022b), https://doi.org/ 10.1594/PANGAEA.950095, (Li et al., 2022c), https://doi.org/10.1594/PANGAEA.950126, (Li et al., 2022d), https://doi. org/10.1594/PANGAEA.950151, (Li et al., 2022e), https://doi.org/10.1594/PANGAEA.950068, (Li et al., 2022f), https: //doi.org/10.1594/PANGAEA.950086, (Li et al., 2022g), https://doi.org/10.1594/PANGAEA.950131, (Li et al., 2022h), https://doi.org/10.1594/PANGAEA.950044, (Li et al., 2022i), https://doi.org/10.1594/PANGAEA.950141, (Li et al., 2022j)

and https://doi.org/10.1594/PANGAEA.950121, (Li et al., 2022k)). Snow pits measurements of density (Paul et al., 2017a), salinity (Paul et al., 2017b) and stratigraphy (Paul et al., 2017c) (https://doi.pangaea.de/10.1594/PANGAEA.881717, https:



**Figure 1.** (a) Map of Antarctica illustrating the geospatial distribution of various in-situ and airborne observation data: OIB (purple circles), ASPeCt (blue rectangles), AWI purchases (yellow stars), IMB (orange crosses), ZS-DS stations (red rhombi), all PS81 ice stations (green triangles), SIPEX II (cyan triangles), and snow observations collected during several Polarstern cruises (pink triangles). (b) A zoomed-in view of the Weddell Sea, showcasing detailed data points. (c) Schematic representation illustrating the functioning of RADIS-L v1.1 in three-layer system: dry snow, brine-wetted snow, and sea ice.





**Figure 2.** Sea ice and atmosphere conditions in (a) AWI 2016S31, (b) PS81/506, (c) PS81/517, and (d) ZS-2010. Sea ice thickness (light red), total ice thickness (snow depth + sea ice thickness) (red) are collected from the buoy measurements. Sea ice concentration (blue) is from AMSR-E/2 datasets, the brine-wetted layer (melt form) percentage (green) is from the SNOWPACK model, and the ice freeboard (green) is from buoy measurements.





**Figure 3.** Observed distribution of (a) snow density (Units: kg m$^{-3}$) and (b) salinity (Units: psu) within different snow stratigraphy in all 13 PS81 ice stations, with the median (white dost), 25th and 75th percentile (black thick vertical bars), whiskers in 95% confidence interval (black thin vertical bars), and all data (points) in the violin. (c) all measured salinity (in orange circles) and density (in purple stars) from 13 ice stations, with their average are in crosses and pluses, respectively. (d) is the frequency (gray bars) and relative snow height (dots) for different snow stratigraphy in all ice stations during PS81 and SIPEX II field campagians.





**Figure 4.** The mean of horizontal and vertical polarised Tbs comparison from SMOS and simulated from RADIS-L v1.0 and RADIS-L v1.1 along the buoy trajectories and scatters fitting in (a) AWI 2016S31, (b) PS81/506, (c) PS81/517, and (d) ZS-2010 buoy.





**Table 1.** Summary of in-situ and satellite data, including parameters used, period and temporal resolution

| Data | Mission/field work | Parameters | Period | Temporal resolution |
|---|---|---|---|---|
| In-situ | ASPeCt | hi, hs, SIC, ice type, $T_{surface}$ | Oct-2010~Dec-2019 | Daily, per cruise |
| | Weddell Sea buoys (PS81/506, PS81/517, 2016S31) | hi, hs, ice type, $T_{surface}$ | Aug-2013~Nov-2013 May-2016~Dec-2016 | Daily, along trajectory |
| | Prydz Bay buoys (ZS-2010, ZS-2013a, ZS-2013b, ZS-2014, ZS-2015, DS-2014, DS-2015, DS-2016, DS-2018a, DS-2018b) | hi, hs | 2010, 2013, 2014, 2015, 2016, 2018 | Daily, along trajectory |
| | Snow pits from Polarstern cruise | snow density, snow salinity | Jun-2013~Aug-2013 | Daily, per pits |
| | OIB | hi, hs, $T_{surface}$ | 26,28,30-Oct-2010 12,25-Oct-2011 20-Oct-2014 27-Oct-2016 | Daily, per campaign |
| | SIPEX II | hi, hs, freeboard, snow density, salinity, morphology | Sep Nov-2012 | Daily, per transect |
| Satellite | SMOS | TB(0-40°) | 2010~2016 | Daily |
| | SMAP | TB(40°) | 2015~2016 | |
| | AMSR-E/AMSR2 | SIC, TB, hs | 2010~2016 | |
| | OSI-SAF | Ice type | 2010~2016 | |
| | ALOS PALSAR | $\sigma^0_{HH}$ | Oct-2010 | |
| Reanalysis | JRA55 | Air temperature, relative humidity, wind speed, precipitation, vertical wind profiles | 2010-2016 | |

//doi.pangaea.de/10.1594/PANGAEA.881714, and https://doi.pangaea.de/10.1594/PANGAEA.881713) are available from ANT-XXII/2 in 2004, ANT-XXIII/7 in 2006, PS89 ANT-XXX/2 in 2013, and PS81 ANT-XXIX/6.

OIB data from NSIDC IceBridge L4 are accessible at: https://nsidc.org/data/IDCSI4/versions/1. L3B Tbs product from
SMOS is available at: https://www.cen.unihamburg.de/en/icdc/data/cryosphere/. Version 2 Tbs from SMAP can be found



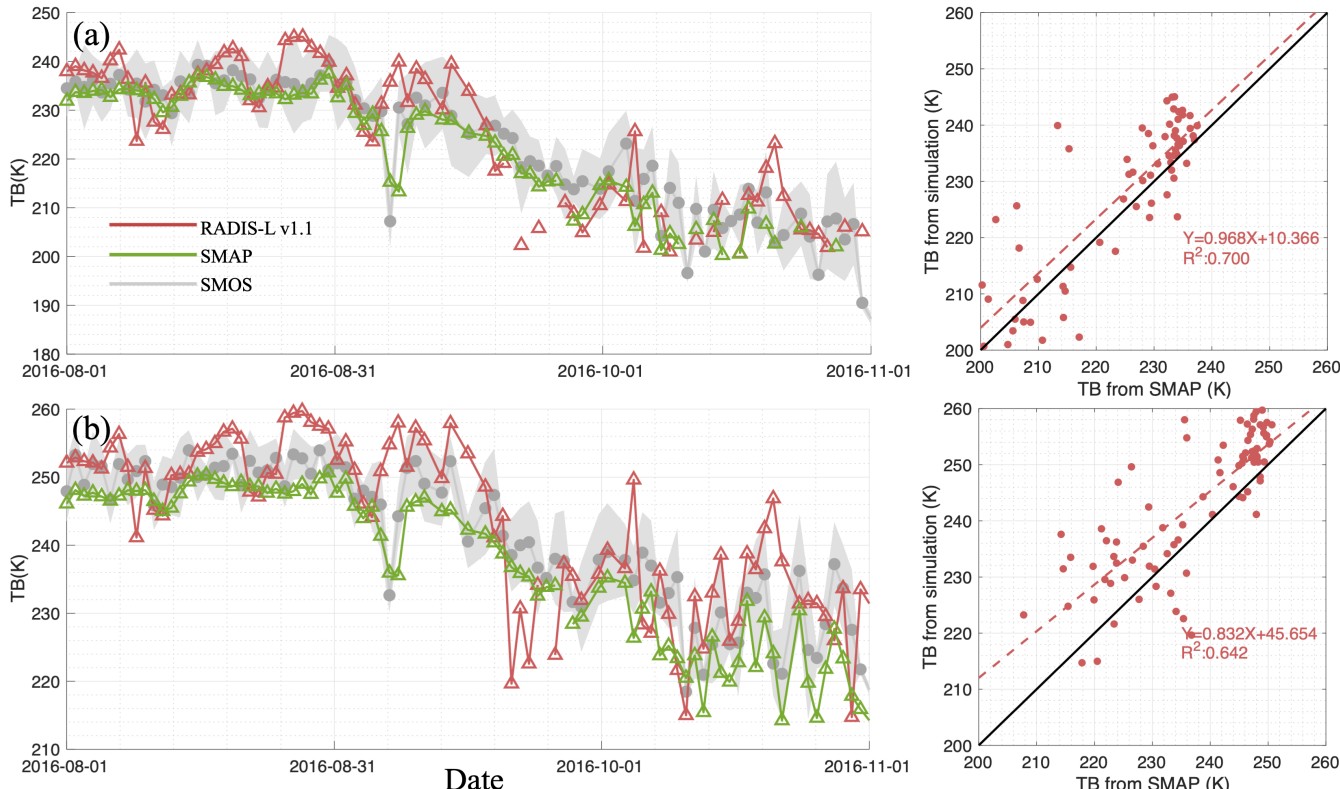

**Figure 5.** The (a) horizontal and (b) vertical Tbs comparison from fixed incidence angle (40°) SMAP, multiple incidence angles (0∼60°) and simulated from RADIS-L v1.1 model along the AWI 2016S31 buoy trajectory and scatters comparison. The shading within SMOS observation is the one standard deviation of multiple angles

at: https://nsidc.org/data/nsidc-0738. Sea ice concentration from AMSR-E/2 is available at: https://nsidc.org/data/AE_ SI12/versions/3. The ALOS PALSAR images are available from the Alaska Satellite Facility (ASF) can be accessed at: https://www.asf.alaska.edu/.





**Figure 6.** The Tbs comparison between RADIS-L v1.1 simulation and SMOS based on (a) ASPeCT measurements. (b) and (c) are the hot map of Tbs validation using all seven OIB campaigns resulting from RADIS-L v1.0 and RADIS-L v1.1. The black dots in (c) are the Tbs overestimation value discussed in 5.1.





**Figure 7.** The distribution of sea ice (red shading), snow (blue shading) and freeboard (green stars) measurement from five transects. The violin shadings cover the range of 1st and 99th percentiles, the upper (lower) boundaries of slim boxes are the mean +(-) standard deviation, while the upper (lower) ones in thick boxes are the 3rd and 1st quartiles of parameters. The short-solid horizontal lines represent the median of observations. Purple stars, upper triagnles, and lower triangles are Tbs from SMOS observations and simulation from RADIS-v1.0 and RADIS-v1.1 with slush, respectively. The crosses are observed air temperature in each station.



**Figure 8.** The differences between simulated and SMOS Tbs within the 28-Oct-2010 track overlaid by sea ice concentration (a) from AMSR-E and by the HH-polarised backscatter from ALOS L-band PALSAR on 30-Oct-2010 ((b) and (c)) and 29-Oct-2010 ((d) and (e)), and the (f) distribution of backscatter during these two days.







**Figure 9.** Three configurations of slush layer with different water content ($\theta_w$, units: %) and air content ($\theta_a$, units: %) in (a), (b), and (c). (d) is simulated Tbs (Units: K) changing with different slush depth (Units: %) under different slush properties (dashed and dotted lines) and percentage of overlaid brine-wetted layer (different colors). Four horizontal lines represent the simulated Tbs in different brine-wetted layer depth without the slush layers.





**Figure 10.** Tbs distribution under the perturbation of (a) and (c) snow density (units: kg m$^{-3}$) and (b) and (d) salinity (units: g kg$^{-1}$, and (e) ((f)) are under the constant (value-increased) ice and snow standard deviation Monte-Carlo perturbation in violin distribution. The stars are the Tbs values in single ice and snow measurements based on RADIS-L v1.1.





**Table A1.** Input parameters and their sources in RADIS-L v1.1 model. OBS is the abbreviation of direct observations

| Validations | hi (m) | hs (m) | Ice type | Surface temperature (K) | Sea ice concentration (%) | Relative brine-wetted depth (%) |
|---|---|---|---|---|---|---|
| Snow buoys | Buoy OBS | Buoy OBS | Buoy OBS | Buoy OBS | ASI | 2016S31 (Wever et al., 2020); PS86/506, PS81/517 (Wever et al., 2021) |
| SIMBA-type buoys | Buoy OBS | Buoy OBS | OSI-SAF | Buoy OBS | ASI | Negative ice freeboard-derived |
| ASPeCt | ASPeCt OBS | ASPeCt OBS | ASPeCt OBS | ASPeCt OBS | ASPeCt OBS | Negative ice freeboard-derived |
| OIB | OIB OBS | OIB OBS | OSI-SAF | OIB OBS | ASI | Negative ice freeboard-derived |
| SIPEX II | Ice Station OBS | Ice Station OBS | OSI-SAF | Ice Station OBS | ASI | Slush depth*: Negative ice freeboard-derived |

* Observed depth, density and salinity of slush from five Ice Stations are used to simulate the Tbs in Sec. 4.3.

## Appendix A: Supplementary table



**Appendix B: Supplementary figures**

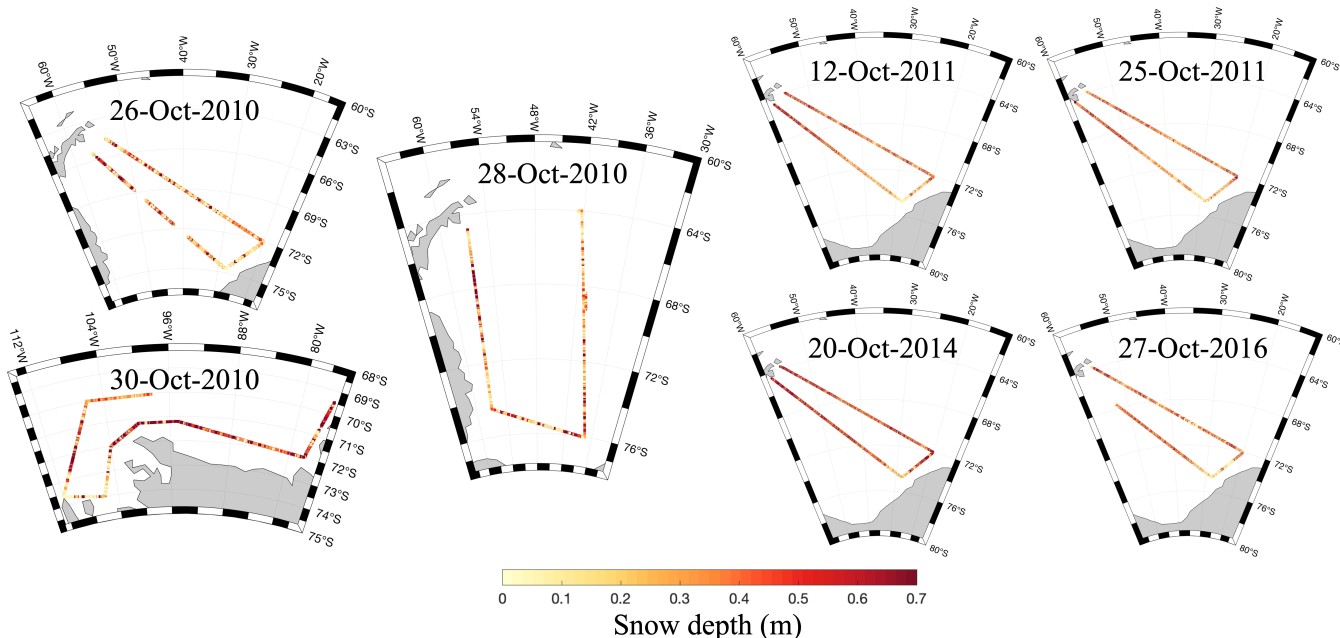

**Figure B.1.** Snow depth (Units: m) retrieved from NSIDC L4 datasets during 2010 OIB campaigns and from average of Wavelet and Peakiness algorithms during 2011-2016 OIB campaigns.

*Author contributions.* LZ and SMX designed and performed the experiments. WXZ and ZFY processed the ALOS and JRA55 datasets. SK performed the OIB datasets. LZ, JS, and RW wrote the majority of the main text. All authors provided insights regarding the interpretation of data and reviewed and edited the manuscript.

*Competing interests.* At least one of the co-authors is the member of Editorial Board of The Cryosphere.

*Acknowledgements.* This work is partially funded by the joint project of INTERAAC co-funded by the National Key Research and Development Program of China (project no. 2022YFE010670) and the Research Council of Norway (grant no. 328957). SX is also partially funded by the National Natural Science Foundation of China (project no. 42030602), the Research Council of Norway through the project of TARDIS (grant no. 325241), and the International Partnership Program of Chinese Academy of Sciences (grant no.: 183311KYSB20200015). LZ is partially funded by the Swedish National Space Agency (grant no.:
164/18). JS is funded by Canada C150 Program (grant no. 50296). JS and RW are funded by European Union's Horizon



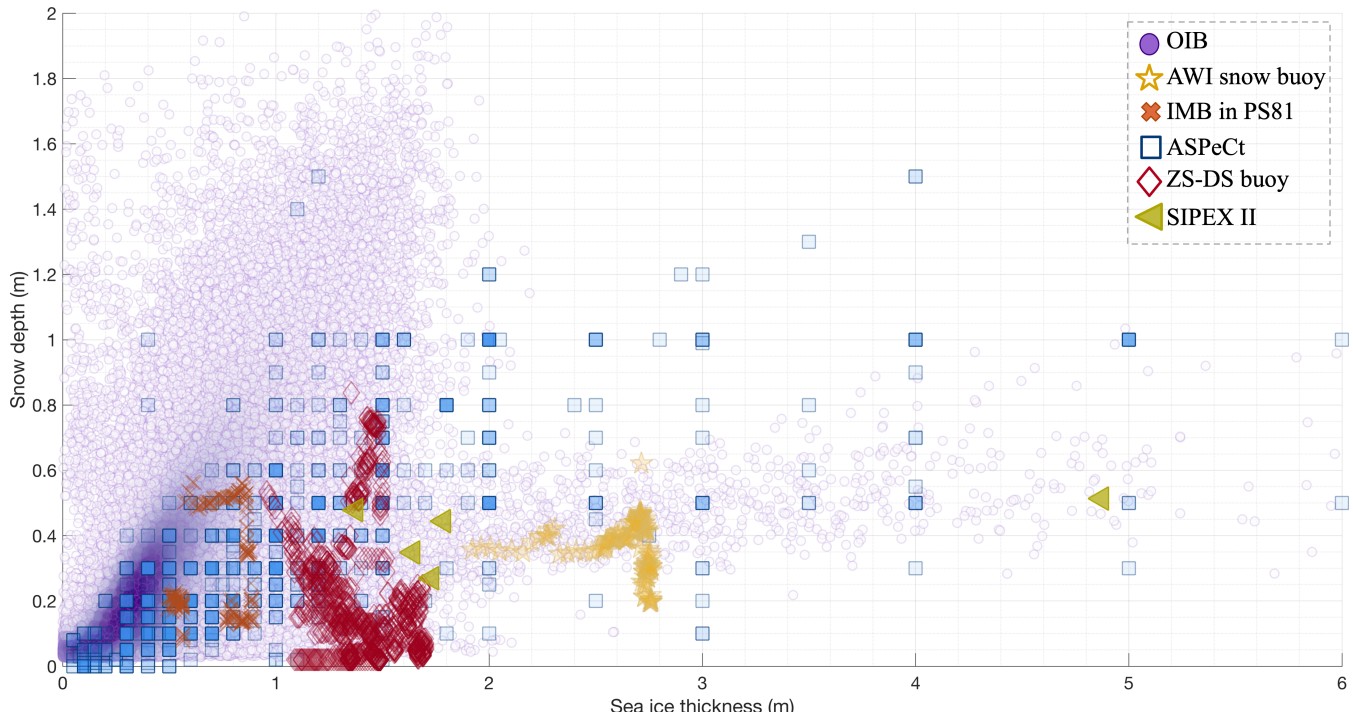

**Figure B.2.** All measured sea ice thickness (Units: m) and snow depth (Units: m) from OIB campaigns, AWI snow buoy, PS81 expedition, and ASPeCt.

2020 LC-CLA-17-2020 CRiceS (grant no.: 101003826) and NERC DEFIANT (NE/W004712/1). SA received funding from the German Research Foundation's (DFG) projects fAntasie (grant no.: AR1236/3-1) and SnowCast (grant no.: AR1236/1-1) within its priority program "Antarctic Research with comparative investigations in the Arctic ice areas" (grant no.: SPP1158), the DFG Emmy Noether Programme project SNOWflAke (grant no.: 493362232) and the Alfred-Wegener-Institut, Helmholtz-Zentrum für Polar- und Meeresforschung. SK performed the OIB work at the Jet Propulsion Laboratory, California Institute of Technology, under contract with the National Aeronautics and Space Administration. We thank Ronald Kwok for providing the Operation IceBridge freeboard data.



**Figure B.3.** Snow stratigraphy modeled from the SNOWPACK in (a) AWI 2016S31, (b) PS81/506, and (c) PS81/517



**Figure B.4.** Observed snow density (Units: kg m$^{-3}$) within different snow stratigraphy in the ice station (a) PS81/506 and (b) PS81/517



**Figure B.5.** Tbs validation between simulation from RADIS-L v1.1 and SMOS based on all 10 SIMBA-types buoys measurements over Prydz Bay.





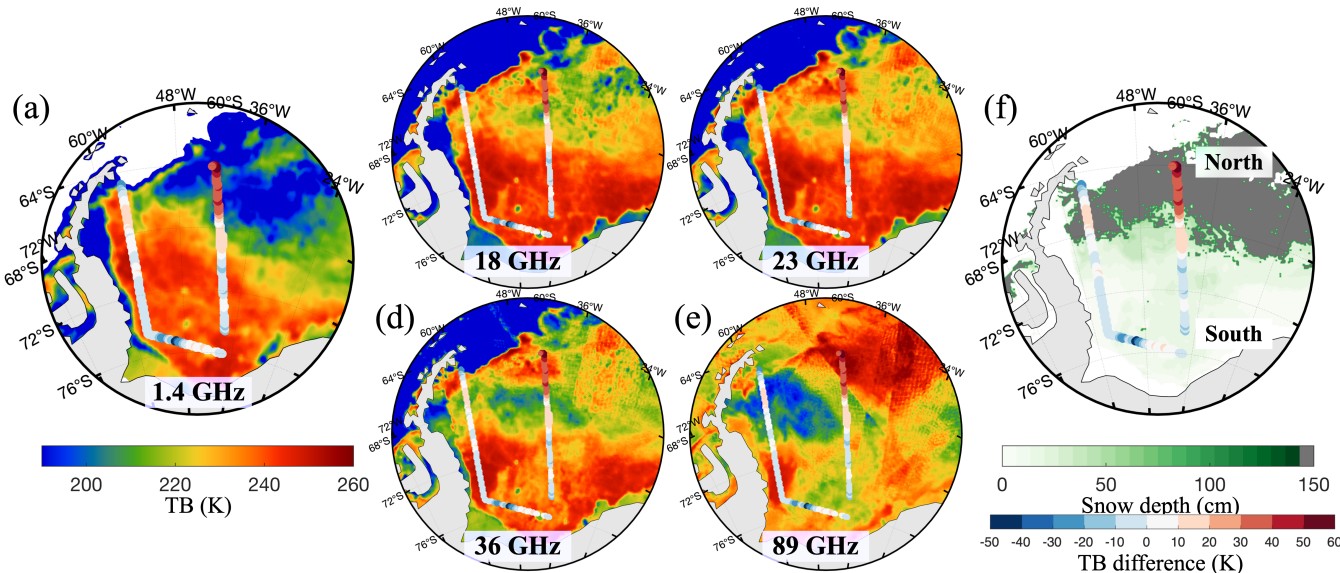

**Figure B.6.** The differences (circles coloured red to blue) between simulated and SMOS Tbs within the 28-Oct-2010 track overlaid by Tbs in (a) SMOS (1.4 GHz) and different AMSR-E frequencies from (18 ∼ 89 GHz), (b) to (e). Snow depth map (f) is obtained from AMSR-E products.

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
