# Peer review of "Quantifying the Influence of Snow over Sea Ice Morphology on L-Band Passive Microwave Satellite Observations in the Southern Ocean"

_EGUsphere, 2024_

## Author Comment (AC1)

The authors would like to thank the editor and the referees for their comments on our manuscript. Following the comments, we make the following replies and corresponding revisions to the manuscript. Each item of the original comments from the referee is in *blue italic*, followed by our reply. Moreover, in the marked version of the revised manuscript, the revisions are highlighted with 'REV1'.

**REVIEW 1**

*Review on "Quantifying the Influence of Snow over Sea Ice Morphology on L-Band Microwave Satellite Observations in the Southern Ocean" by*

*Zhou et al.*

*Utilizing the limited but widely collected data for the sea ice and snow in the south ocean by buoys, ship bases, and aviation satellites, based on the snowpack and radiation transfer models, the authors simulated and analyzed the impact of snow cover and its stratification on the passive microwave brightness temperature observation results. The latter is crucial for satellite remote sensing inversion of sea ice concentration and snow/sea ice thickness. Overall, the authors conducted a very detailed analysis based on the complex data. The comprehensiveness of the data is the greatest contribution of the paper, and the complexity of the data also brings many uncertainties to the results of the paper. Therefore, I recommend that the paper needs to undergo major revisions before it can be considered for publication. My main concern is still to focus on the analysis of complex data. Here are my general comments:*

- *Does the snowpackmodel consider the effect of snowdrift? Is the thickness of snow simulated output or forced by observational data?*

**Reply**: we would like to clarify that: (1) SNOWPACK model does not include snow drift processes, and (2) we are aware of its limitations, we only use SNOWPACK in specific ways. For AWI buoys which do not directly report inundation, for the simulation of Tbs in RADIS-L v1.1, we use the percentage of this brine-wetted snow layer reported by SNOWPACK. For comparison, for cases which directly report negative freeboard (e.g., ZS-2010), the direct measurements are used instead.

Our second clarification is: the snow depth from direct observations (e.g., SIMBA buoys, OIB) are used to force the RADIS-L radiation model.

- *How are the differences in snow and sea ice characteristics in different regions, especially between the regions of drifting ice and landfast ice, considered?*

**Reply**: we agree with the referee that this is important, and are also fully aware of the fact that vast differences in snow and ice properties are present for the various observations used in this study. On the other hand, both the radiation model we develop (i.e., RADIS-L) and the SNOWPACK that we utilize are general-purpose tools that accommodate these large variability of key sea ice parameters such as snow and ice

thickness, as well as snow-ice formation due to inundation. We acknowledge that certain limitations exist, such as the different levels of ridging between sea ice in Weddell Sea and other places. They may constitute sources of uncertainty when comparing data of different origins (e.g. F.g. 4-6). However, as shown in this study, in spite of the large variability of the data sources and conditions, the snow-ice formation and the effect on L-band Tbs is supported in in-situ measurements, and evidently, much better agreement with satellite Tbs. We fully agree that environmental and representation factors need to be considered fully, which we have planned for future work.

*Special comments:*

- *Line 18 "facilitating key biological processes"is not belong to the influences on the polar climate system"*

**Reply**: we revise the whole sentence as follows: "Snow on sea ice significantly influences the polar climate and ecosystems by mediating mass and energy exchanges during air-sea interactions, as well as key biological and biogeochemical processes".

- *Line 57: Why the evolving climate conditions in Antarctica would lead to snow melting and refreezing processes becoming more prevalent?*

**Reply**: we would like to clarify that: with warming of ocean on lower latitudes and frequent storm intrusion, the relatively warm surface conditions (even above 0 degC) are becoming more frequent over the sea ice. Hence the melt-refreeze cycles both over the top and bottom of the sea ice are more prevalent.

- *Line 76 "vis-à-vis"it is not a commonly used word.*

**Reply**: we have changed "vis-à-vis" to "with regard to", which is more commonly used.

- *Line 94 "We also use data collected from ice mass balance buoys (IMBs)": I wonder if this is also SIMBA buoys, as it is also not equipped with sonars. Please verify.*

**Reply**: we have verified that the buoys deployed are of SIMBA type (Wever et al., 2021). Therefore the manuscript is revised as: "*We also use data collected from ice mass balance buoys (IMBs) and snow buoys in the Weddell Sea from 2013-2014, and 2016. They are the SIMBA type buoys and equipped with a thermistor string to measure sea ice temperatures with a vertical spacing of 0.02 m.*"

Since in Wever et al. (2021), the abbreviation of IMB (for Ice Mass Balance buoy) is used for SIMBA-type buoy as well, we choose to follow this practice in this article.

- *SIMBA-type buoy over Prydz Bay: the buoys deployed in the Prydz Bay included the SIMBA and CRREL-IMB.*

**Reply**: we thank the referee for pointing out this inaccurate statement. We have revised it to "Ice mass balance buoys over Prydz Bay".

- *Line 127 "Vertical now salinity profiles": It is snow profiles.*

**Reply**: corrected.

- *Line 153 " et al. (2015)": typing error.*

**Reply**: corrected.

- *ASPeCt snow and ice thickness data: This data actually has a significant error because it is visually judged by humans.*

**Reply**: we agree with the referee that ASPeCt thickness data contain relatively larger uncertainty than other in-situ data we use (e.g., buoys). Besides, similar representation error is also present for ASPeCt. However, it provides us with an important source of information, since observations are scarce for sea ice in Southern Oceans. Beyond this study, ASPeCt are also frequently used in many studies for the validation of satellite retrieval of sea ice thickness.

- *Method by Cox and Weeks (1983) and Leppäranta and Manninen (1988) is used to calculate the volume fraction of brine inside sea ice. Whether it is also suitable for the slush layer?*

**Reply**: we consider the brine volume fraction estimation in Cox & Weeks (1983) is sufficient for this study, which allows large and realistic brine volume under relatively warm conditions (i.e., near freezing). For example, the brine volume fraction at -2degC and 0.8m ice is 0.151.

- *Snow evolution off Zhongshan and Davis Stations in Prydz Bay: According to observations, negative ice freeboards and slush layers are rare in this regions because of strong katabatic downwind. See also:*
- *Li N, Lei R, Heil P, Cheng B, Ding M, Tian Z, and Li B. 2023.Seasonal and interannual variability of the landfast ice mass balance between 2009 and 2018 in Prydz Bay, East Antarctica, The Cryosphere, 17, 917–937, https://doi.org/10.5194/tc-17-917-2023.*
- *Lei R, Li Z, Cheng B, Zhang Z, Heil P. 2010. Annual cycle of landfast sea ice in Prydz Bay,east Antarctica. Journal of Geophysical Research: Oceans, 115(C2), C02006, 1–*

*Thus, is the data of the ZS-2010 buoy representative? This buoy observed a negative ice freeboard because it was deployed near an iceberg. In addition, in the observation footprints of satellite remote sensing, the vicinity of this buoy should include signals from icebergs and land.*

**Reply**: we consider the buoy ZS-2010 to be representative of the snow-ice formation and the change in Tbs. As pointed out by the referee, ZS-2010 resides on landfast sea ice for the period of study, and the surrounding conditions such as the iceberg remain the same.

According to the suggestion of the referee, we have revised the statements on line 360 that imply the causality of snow accumulation on snow-ice formation at ZS-2010 (i.e., the potential role of the nearby iceberg on the freeboard).

- *Line 405 "Essentially, this means that the conditions observed at the buoy location are representative of the entire grid cell, ensuring that the satellite Tb data is a valid proxy for the conditions across the whole floe"*

*In fact, the surface brightness temperature at the floe scale depends not only on air temperature and lead distribution (or ice concentration), but also on the heterogeneity of snow and sea ice thickness, the latter of which undergoes significant changes at the floe scale of several tens of meters.*

**Reply**: we agree with the referee that the buoys have potential representation issues. We would like to clarify that the word "means" here should be "implies". It is revised accordingly.

- *Line 458 "we conduct Tbs simulations following the trajectory of the ZS-2010 buoy"*

*In fact, this buoy is deployed on landfast ice and would not drift, with the small distance from the shore.*

**Reply**: we have revised it to "we conduct Tbs simulations at the location of the ZS-2010 buoy".

- *Figs. 4-6: Do you also consider using surface temperature observed by buoys to compare/verify brightness temperature?*

**Reply**: we confirm that the brightness temperatures are simulated using: (i) buoy measurements when temperature data is available. (ii) JRA55 2m temperature data when buoy temperature data is missing. These methods are used for buoy-based validations (see Tab. A1). In fact, we have conducted a validation comparing the temperatures between the buoy (represented in light purple) and JRA55 2m (represented in dark purple), as shown in the figure below. The correlation in air surface temperature between the buoy and JRA55 2m data is over 0.95, indicating a strong agreement with no significant bias.

---

## Author Comment (AC2)

The authors would like to thank the editor and the referee's comments on our manuscript. Following the comments, we make the following replies and corresponding revisions to the manuscript. Each item of the original comments from the referee is in *red italic*, followed by our reply. Moreover, in the marked version of the revised manuscript, the revisions are highlighted with 'REV3'.

**REVIEW 3**

*Summary*

*Flooding and slush over Antarctic sea ice have been a major reason to cause errors and uncertainty for snow depth and/or sea ice thickness retrieval from Satellite remote sensing for southern oceans. The purpose of the paper hopes to improve snow depth and ice thickness retrieval from passive microwave (1.4 GHz) brightness temperature via satellites such as SMOS and SMAP, by improving the existing RADIS-L v1.0 to v1.1, by introducing brine-wetted snow and lush layers over Antarctic sea ice. The modeling results are validated based on three types of datasets: 4 buoys, some ASPeCT data, and OIB data. Very surprisingly, the ASPeCT data actually gave the best results, only 1 out of 4 buoys gave reasonable results, validation from OIB seems not as good as stated. They attribute the biases between modeled and observed Tbs mostly to sub-grid scale ice surface variability and snow stratigraphy. One of the results is that the adding of a slush layer into the model decrease the modeled Tbs. This is obviously true. Anyway, I agree this is very important topic of study, but the paper needs to prove that the v1.1 is indeed effectively modeling the wet and slush layers for Tbs, and in my opinion, it is not the case in its current format.*

*General comments*

*The paper in somehow confuses me on the brine-wetted and slush layers. I hope they can make corrections throughout the paper, to brine-wetted snow layer and slush layer, since slush layer is not snow layer any more. Slush layer is a mixed snow, ice, and water, more close to water in my opinion. And the slush layer does not exist too long (not sure how long, may be a few hours?) before refreezing to become the ice (or snow-ice).*

**Reply**: we agree with the referee's suggestion to clearly distinguish the brine-wetted snow and slush. For this study, the highest temporal scale is 1-day (Table 1), especially the comparison of modeled Tbs with satellite measurements. Therefore, we consider the slush due to inundation an intermediate stage, and we mainly focus on snow-ice, which may also contain a significant portion of brine (i.e. liquid content). Necessary revisions are made to the manuscript to be more clear about several concepts: slush, brine-wetted snow, and snow-ice.

*Also need to make sure the improvement of the modeling from v1.0 to 1.1 is only adding the "slush layer", since the v1.0 already has the brine-wetted snow layer. Right?*

**Reply**: the v1.0 of RADIS-L does not include slush or snow-ice, since the model was developed for the Arctic region, where snow-ice is not a dominant feature. The new version of RADIS-L (i.e., version 1.1) adds the support of snow-ice and its effect on Tbs.

*The paper in somehow confuses me about active and passive microwave sensor. From the title, I thought it uses radar data, but clearly the paper did not use radar data. see my comment on L166, I suggest to change "L-band" to either "passive" or "1.4 GHz". If this is agreed, you should do it throughout the entire paper.*

**Reply**: we confirm that the L-band passive microwave remote sensing is the focus of the study. We also use ALOS PALSAR (i.e., L-band SAR) images (HH-polarization) as auxiliary data to the result analysis.

As suggested by the referee, we slightly change the title to: "Quantifying the Influence of Snow over Sea Ice Morphology on L-Band Passive Microwave Satellite Observations in the Southern Ocean".

*Specific comments*

*Line 7, change to "brine-wetted snow and slush layers"*

**Reply**: revised accordingly.

*L35, should be "zero ice freeboard"*

**Reply**: revised accordingly.

*L127, do you mean "vertical snow salinity"?*

**Reply**: corrected.

*L166, "this fully polarized sensor…." is wrong. I do not think it is fully polarized. Since SMOS is passive and it is either H or V, not HH, HV, VH, or HH. Also my comment about the title. L-band is mostly for active sensor, for passive sensor, we usually use frequency.*

**Reply**: we acknowledge this incorrect statement. This sentence is revised as: "This sensor measures the Earth's emitted radiation at 1.4 GHz at both horizontal and vertical polarization and multiple incidence angles".

*L199, should be "horizontal transmit and receive polarization", not the other way as here. Remember if it is active, it transmits first, then receives.*

**Reply**: revised according to the referee's comment.

*L204, I do not know why you want to use 35 degree, not the real incidence angle for each pixel? I think SNAP can do it by directly use the real incidence angle of each pixel.*

**Reply**: we choose to correct for the incidence angle the PALSAR image to 35-deg, in order to get a more uniform image. This image is then used for visual inspection for key characteristics such as sea ice concentration. The use of the un-corrected, original image (from SNAP) is possible, but not necessary.

*Section 3.1 SNOWPACK. Is this RADIS-L v1.0? if yes, should you change the section to it? If not, what it is the difference between them?*

**Reply**: we would like to clarify that SNOWPACK and RADIS-L are two different models. SNOWPACK (Wever et al., 2020) is provided by third-party developers, and it is a prognostic snow-sea ice model for snow stratigraphy and forced with atmospheric forcings. We use SNOWPACK to simulate possible stratigraphy features and layering of the snow pack over sea ice.

On the other hand, RADIS-L is developed by the authors, and it is a radiation model for L-band radiometers. The improvement of RADIS-L (from v1.0 to v1.1) mainly involves the inclusion of slush/snow-ice.

*L243, should "X- and L-band" be the relevant frequency of them?*

**Reply**: we confirm that the two bands of L-band and X-band were the original focus of the RADIS-L model for passive radiometers.

*L282, equation 4, the surface Temperature Tsurf, I believe it is more controlled by the air temperature, rather than as provided by the equation.*

**Reply**: we would like to clarify that we do NOT mean that Tsurf is controlled by the equation. Instead, under quasi-equilibrium status for themodynamics, the temperature profile from the surface (i.e., Tsurf) to the internal (i.e., snow-ice interface) follows the relationship in Eqs. 4.

*L305, "Therefore, we treat the snow slush …..", please check this sentence and make sure it is the right statement? I am little confused.*

**Reply**: according to the suggestion by the referee, we revise the sentence as: "Therefore, we simply treat the slush as newly formed sea ice, with a variable and high volume of brine".

*Section 3.3, wondering why you want to use 30-50km, not use the 9km for SMAP and 12.5 km for SMOS?*

**Reply**: we would like to clarify that the effective resolution of both SMOS and SMAP is in the range of 35-50 km. The enhanced resolutions of 9 km and 12.5 km are achieved using nearest-neighbor algorithm interpolation. This method reduces geophysical and temporal variability and brightness temperature uncertainties by aggregating single-angle brightness temperature measurements for each grid point. The process of improving the resolution from coarse (e.g., 35-50 km) to fine (e.g., 9 or 12.5 km) using interpolation, along with the characteristics of L-band data, introduces limitations. These factors can affect the accuracy of the fine resolution data and result in nearby data points being correlated rather than truly independent measurements. Therefore, we choose the coarse resolution to adhere to the band properties and minimize influences from nearby grid cells. For more details, please refer to the L3B brightness temperature (Kaleschke et al., 2012) data guide: https://www.cen.uni-hamburg.de/en/icdc/data/cryosphere/docs-cryo/documentation-smos-tbs.pdf

*L353-354, when there was heavy snowfall, it will general flooding, slush, then slush freezing to snow-ice; this entire process should increase ice thickness not decrease ice thickness. I do not understand your statement here.*

**Reply**: we hereby correct the previous wrong statement. And as mentioned by the referee, the heavy snowfall could potentially increase the ice growth. And according to buoy measurement, the sentence is revised into "Snow and ice remain stationary, followed by heavy snowfall (over 35 cm snow depth) starting from 11-Sep-2013 in PS81/506 (Fig. 2b)."

*L402-406, this is hard to align the satellite pixel with the buoy location, given the buoy collecting continuous data while the satellite only gets one data per day. Also the buoy (ice floe) is moving. Not sure how you process your buoy data to match the satellite data? You average all data for the day or you only use the data collected when the satellite passing? Maybe averaging these data a few hours before and after the satellite passing?*

**Reply**: we would like to clarify that all data are aligned to a daily basis. First, the Tb product (SMOS) has relatively coarse spatial resolution, although more than 1 satellite pass is possible within a day. Therefore the daily average (of several passes) is computed for Tb. Second, we compute the daily mean location of the buoy and find the cell of the Tb product that contains the buoy's (mean) location. Due to the relative coarse resolution of Tb, we do not encounter problems due to the buoy's drift (i.e., no case with the buoy covering a large number of cells within a day).

*L407-408, this is a good statement but it is only true for 1 out of 4 buoys. See figure 4 (a). this makes me to wonder something is wrong. This would be due to my comment on the line 402-406, or other reasons we do not know. I believe you need far more example than these 4 and may be also expand to other sections of the Antarctic, currently 3 in Weddell sea. The one not in Weddell sea is a fast ice which could be very different from the pack ice situation. And I am not sure if the flooding and slush layer would happen at all for fast ice case. I take it back, from figure 2d, it shows negative freeboard. But please check if this is indeed.*

**Reply**: we would like to clarify that as referee 3 mentioned, the negative ice freeboard was observed over the ZS-2010 buoy, and referee 1 also mentioned that the negative ice freeboard occurred near the iceberg and substantial snow accumulation directly contributed to the snow-ice formation. While the simulation accuracy of Tb might be limited due to the presence of the iceberg, any changes in Tb are likely attributed to variations in snow since the iceberg remains constant. Secondly, We also want to emphasize that the inclusion of brine-wetted snow improves the simulation of Tb. Significant improvements were observed for buoys 1, 2, and 4. Although the improvement for buoy 3 is small in terms of the statistical fitting and $R^2$, the overestimation of Tb caused by snow did witness a reduction. We believe the improved model better captures the changes in Tb due to sudden daily snow accumulation as seen in buoys 1, 3, and 4, though further progress is needed for systems with existing brine-wetted snow. Lastly, we want to note that the buoy-based data used in this paper, which contain both snow and ice measurements, already represent the most comprehensive dataset available to date. We would be eager to validate our findings with additional data if new data becomes available.

*L415-416. It is true "Tbs reduction from 243.8k ….to 226.1K",  but it also involves with a few increases and decreases in between. Can you explain that.*

**Reply**: we would like to clarify that the variability of modeled Tb is largely due to that of the surface temperature as measured by buoys (in Fig. 4). The buoy's Tsurf contains large temporal changes (i.e., decreases and increases) due to atmospheric activities. Since we assume thermal quasi-equilibrium, this translates to the large changes of internal temperature as well (and Tbs). We argue that although this assumption leads to artificially large variability of internal temperatures, the long-term (beyond several days) change of Tbs in deed contains the physical signal of atmospheric signature of cooling and warming.

*L458, if ZS-2010 buoy was on the fastice, why you talk about trajectory? Please explain here.*

**Reply**: we have revised the sentence to: "... we conduct Tbs simulations at the location of the ZS-2010 buoy".

*L462-467 and figure 4d, your statements seem be suspicious. I see v1.0 and 1.1 have the same patterns but different magnitudes. Compare with satellite Tbs, however, there are a few cases, with inverse change. R2 of 0.35 does not mean a good relation, in my opinion.*

**Reply**: As mentioned earlier, the presence of a nearby iceberg and proximity to the coast (Li et al., 2023, see their Figure 1) affect the Tb measurements from SMOS. The emissivity from the iceberg and ice shelf within 30-50 km influences these measurements (see the buoy location in Fig.1, overlaid by Tb on September 1, 2010). Consequently, it is a unique study case which is different from the ones on drifting buoys in the Weddell Sea. Nonetheless, most of the Tb variability due to significant snow accumulation is captured, with the exception of the abrupt increases and decreases around October 25th. We have confirmed that this is due to a sudden change in the sea ice concentration (light blue line in Fig.2).

[Figure]

Fig.1 ZS buoy location overlaid by Tb on September 1, 2010

[Figure]

Fig.2 JRA55 air temperature, sea ice concentration and brightness temperature along ZS-2010 buoy.

*Section 4.2.2, validation with ASPeCT shows better results than the buoys. In line 488-489, please explain to me what you mean small-scale? What kind of parameters in small-scale would increase your modeling results? In term of SAR imagery, it will not give Tbs.*

**Reply**: We would like to clarify that the small-scale features in lines 488-489 pertain to the context of airborne (OIB) validation. In this context, one of the key parameters in the radiation model—sea ice concentration—should be provided at the highest possible resolution. The ASI sea ice concentration retrieved from passive microwave data is currently the highest available resolution at 6.25 km. However, as noted in Section 5.1, SAR images reveal small-scale leads, which are not captured by the ASI sea ice concentration due to its coarser resolution. This inability to differentiate sub-grid scale (small-scale) variabilities is a primary reason for the overestimation of Tbs. Future work using SAR-based sea ice concentration, which can provide sea ice concentration data at resolutions finer than 1 km, could help reduce this bias and improve the accuracy of the radiation model.

*L492-493, not sure what you mean "This analysis is …. Near Wilkes Land"?*

**Reply**: in order to be more clear, we revise the sentence to: "This analysis is based on observations from five snow and ice transects located near Wilkes Land". We have also added the reference to Fig. 1 in which the locations of the campaigns are shown (see the next comment).

*L495-497, not sure where you mentioned ice stations 2,3,4,6,7? They are in the figure 1?*

**Reply**: the locations of the SIPEX II campaign (the ice stations) are shown in Fig. 1. Necessary reference is added.

*L565-569, mentioned about 2.5m ice and 0.5 m snow for the experiment, I am not sure if this snow depth would cause flooding and slush over ice? It may be but I am not sure. but figure 10 shows 0.2m snow with 2m ice, 0.3m snow with 3m ice, etc.., I am pretty sure these cases would not cause flooding and slush…; please explain. Also as you know and talked about that most of the Antarctic sea ice is thin ice. Your ice here are 2-6m, which is very unnormal. I would suggest to model ice thickness from 0.5 to 3m, the maximum.*

**Reply**: We acknowledge that the combination of sea ice and snow thickness cannot trigger flooding under the assumption of hydrostatic condition. As suggested by the referee, we have revised the sensitivity tests in the updated version of the manuscript.. Specifically, we have adjusted Figure 10 to include sea ice thickness values of 0.5, 1, 1.5, 2, 2.5, and 3 meters, and correspondingly snow depths of 0.2, 0.35, 0.5, 0.6, 0.75, and 0.8 meters, so as to maintain negative freeboard. The sensitivities are updated below:

[Figure]

*L657-658, "Given these considerations…", this statement is very interesting, since from your results, the ASPeCt data seems gave the best validate of simulation. Please explain.*

**Reply**: As noted by Referee 1, ASPeCt data has significant uncertainty due to its reliance on visual judgments made by humans. Therefore, we emphasize the need for caution regarding these uncertainties associated with ship-based sea ice and snow measurements.

*Conclusions section includes a lot of contents that are not really from this paper, many of them could be included in the Discussion section, some of others are very suspicious*

*statements. For example, line 642-643, I do not see where you discussed this and how you come to this: "The scenarios we have discussed …in various settings".*

**Reply**: we have revised the paper, to include the discussion in Sec. 6 (Conclusions) to Sec. 5 (Discussion). Also, as suggested by the referee, we refrain from claiming that "The scenarios we have discussed provide critical benchmarks for improving radiometer designs…". Careful examinations are also carried out to improve the soundness of the this part of the manuscript.

*Figure 1. why you only show three layers, not the fourth layers that you claimed to improve in the v1.1? .To me this is for the v1.0, right?*

**Reply**: we want to clarify that Fig. 1 is for v1.1, which is designed as a simple and standard Three-layers scheme: dry snow, brine-wetted snow, and sea ice. Four-layers scheme (dry snow, brine-wetted snow, slush (snow-ice), and sea ice) is further discussed in Sec. 5.2 to further improve our understanding, and this needs more (slush) information in the Four-layers scheme.

*Figure 4, very similar pattern between v1.0 and 1.1, but really not that match with the observations, except for the case in (a), i.e., the buy 2016S31.*

**Reply**: we want to clarify that from v1.0 to v1.1, the main difference is in areas where brine-wetted snow exists, resulting in a consistent pattern between the two versions but with generally lower Tbs in v1.1. Additionally, the model uses some bulk parameters, and the input datasets have potential representation related uncertainties. Therefore, we expect some discrepancies when compared to the coarse satellite footprint of SMOS.

*Figure 8, I do not see "the differences between simulated and SMOS Tbs …", where are those Tbs in this figure?*

**Reply**: all the pentagram symbols (in panel a, b, c, d, e) are colored according to the difference in Tb (simulated minus observed). Faint colors indicate smaller Tb differences.

*Figure 10. I hope to see the same value range of 242-256 for all y-axis of a, b, c, d.*

**Reply**: according to the suggestion, we have revised Fig. 10 to align the range of the y-axis of the panels.

**Reference**

Li, N., Lei, R., Heil, P., Cheng, B., Ding, M., Tian, Z., and Li, B.: Seasonal and interannual variability of the landfast ice mass balance between 2009 and 2018 in Prydz Bay, East Antarctica, The Cryosphere, 17, 917–937, https://doi.org/10.5194/tc-17-917-2023, 2023.

Kaleschke, L., Tian-Kunze, X., Maaß, N., Mäkynen, M., and Drusch, M.: Sea ice thickness retrieval from SMOS brightness temperatures during the Arctic freeze-up period, Geophysical Research Letters, 39, https://doi.org/10.1029/2012GL050916, 2012.

---

## Author Comment (AC3)

The authors would like to thank the editor and the referee's comments on our manuscript. Following the comments, we make the following replies and corresponding revisions to the manuscript. Each item of the original comments from the referee is in green italic, followed by our reply. Moreover, in the marked version of the revised manuscript, the revisions are highlighted with 'REV2'.

**REVIEW 2**

*Review of*

*Quantifying the Influence of Snow over Sea Ice Morphology on L-Band Microwave Satellite Observations in the Southern Ocean*

*by*

*Zhou, L., et al.*

*Summary*

*This very nice manuscript takes a look at the sensitivity of L-Band brightness temperatures as observed by satellite sensors such as SMAP and SMOS (and in the future CIMR) to snow properties on Antarctic sea ice. The focus lies on changes in the snow properties with respect to processes involved in sea-water entering the ice-snow interface, causing basal snow layers to become moist or wet, and saline, being a precursor for a slush present at the ice-snow interface and subsequent snow-ice formation. Since such L-Band radiometer measurements have been used for sea ice thickness retrieval of sea ice below about 70 cm to 100 cm thick in the Arctic, bearing also potential to estimate the thickness of the snow cover on sea ice, such work is really important. The authors used an impressive suite of various independent observations (in-situ and air-borne) to support their investigations based on numerical modeling of sea ice / snow properties using SNOWPACK and radiative transfer modeling using RADIS-L in two different versions. The manuscript is in general well written and provides a red line to follow through the comparably complex modeling tasks carried out. The presentation and interpretation of the results is well organized and bear sufficiently novel information to warrant publication.*

*I have three general comments where I am convinced the authors need to work on before final publication, though. The authors will also find a suite of specific comments and quite a number of editoral comments and suggestions.*

*General comments:*

*GC1: A lot of your model simulations are based on sea ice with a thickness of 2.5 m. Also the sensitivity studies carried out are mainly based on thicker ice categories. I was wondering how representative your results are in the context of Antarctic sea ice usually*

*being considerably thinner than this value. Typical values for most of the seasonal ice cover (e.g. at least 80% of the annual sea ice cover) are between 0.5 and 1.5 m, aren't they?*

**Reply**: We acknowledge the under-representation of the ice thickness and snow depth choices. Therefore, in the slush analysis (Sec. 5.2) and sensitivity section (Sec. 5.3), we re-selected the ice and snow scenarios to better address this issue. In Sec. 5.3, six scenarios were selected, as shown in the following, with sea ice thickness and snow depth combinations as follows: (0.5m, 0.2m), (1m, 0.35m), (1.5m, 0.5m), (2m, 0.6m), (2.5m, 0.75m), and (3m, 0.8m) to keep the sea ice freeboard around or less than 0. In Sec. 5.2, the slush (considered a stage of snow-ice, and thereafter referred to as snow-ice) parameterization scenario was set to a mid-scenario with a sea ice thickness of 2m and a snow depth of 0.6m. The figures and analysis have been updated accordingly in the manuscript.

[Figure]

*GC2: I am not sure I can follow well enough your motivation for the density and salinity values chosen for the slush layer. I find these not overly "slushy". I see slush as a very wet, more or less saturated layer of a mixture of ice crystals, liquid water and brine - with almost no air - if at all. Hence I would assume the density is as least as large as the one of first-year ice - simply because of the high content of liquid water. I (and the reader) shall be grateful to learn from you, why you considerably divert from this slush concept.*

**Reply**: We agree with the referee that snow-ice typically refers to a mixture of ice crystals, liquid water, and brine. As mentioned by the referee and in Massom et al. (2001), the resultant slush quickly freezes to become meteoric ice or "snow-ice." Therefore, in Sec. 3.2.2 and Sec. 5.2, we treat frozen slush as a stage of (hence a type of) snow-ice. This form of ice is distinguishable from other types of granular ice (e.g., frazil ice) and snow morphology due to its coarser grains and higher air bubble content. Snow-ice can be either fresh or saline depending on the conditions of saturation. As stated in Jutras et al. (2016), the salinity of new sea ice immediately after flooding exceeds 20 g/kg. This is consistent with the salinity range of 4.7 to 34.8 g/kg for the snow basal layer around Antarctic sea ice (Massom et al., 1998). Therefore, we will use

20 g/kg as the salinity for snow-ice. Following Saloranta et al. (2000), the physical properties of snow-ice are determined to be: $\rho_{snow-ice}$ = 875 kg/m$^3$, thus snow-ice conductivity ($K_{snow-ice}$) bulk value is 1.045 Wm$^{-1}$K$^{-1}$ based on Eq. 11 in the manuscripts.

*GC3: My impression is that quite a number of the causalities that are presented but also of the explanations given in the context of interpreting the results would benefit from revisiting the physical principles and taking another look at existing literature to sharpen the formulations and avoid misunderstandings.*

**Reply**: we would like to thank to the referee for the suggestions. And as the below detailed comments from the referee, we've added extra physical mechanism and explanation in some specific comments in the following.

*Specific comments:*

*L1: "refrozen snow-ice" --> "snow ice", because this is per definitionem frozen, right?*

**Reply**: revised.

*L26: "thin ice" --> I suggest to drop the "thin" here as it can be mistaken as the WMO's classification where thin ice is sea ice of less than 30 cm thickness.*

**Reply**: revised.

*L28: Again you use the term "refrozen snow-ice". My understanding is that snow-ice is refrozen slush that once formed during a flooding process. Snow-ice is per se frozen - otherwise it would not be termed ice. I therefore suggest to drop "refrozen" whereever you write about "snow-ice"*

**Reply**: revised, all other cases of "refrozen snow-ice" are corrected as well.

*L30-32: "this flooding .... on the top" --> I suggest to sharpen this statement. Is flooding just "accompanied" by the processes mentioned? Or are these processes perhaps rather a pre-requisite for flooding to occur?*

**Reply**: we revise the sentence by separating it into two: "... - this flooding is often accompanied by high ocean heat flux melting from the bottom and redistribution and precipitation of snow occurring on the top" revised into: "This flooding is pre-conditioned

by high ocean heat flux melting ice from the bottom and/or snow redistribution and precipitation on top, which can lower the snow-ice interface below sea level."

*- I suggest to add "of sea ice" behind "melting"*

**Reply**: pls see the last revision item.

*- I suggest to change "redistribution and precipitation of snow" into "precipitation of snow and its redistribution by winds"*

**Reply**: pls see the last revision item.

*L33: "roughly one-third" -- I am doubting that we already have a circum-Antarctic quantification of the amount of snow-ice based on observations. I therefore suggest to be a bit more careful here and write "up to one third".*

**Reply**: revised to the more proper statement of "up to one third".

*L35: "will reassert the balance and produce a zero freeboard"*

*i) Which balance? Please be more specific.*

**Reply**: "hydrostatic" is added before "balance".

*ii) The freeboard you are refering to here is the "sea ice freeboard" and you should write it as such to distinguish it from the total (sea ice + snow) freebaord.*

**Reply**: "sea ice" is added before "freeboard". All other cases are also revised in the manuscript.

*iii) I am not sure I buy the "zero" because slush might be spread on top of the sea ice both below and above the water line and the refreezing of it might turn the entire slush layer into snow-ice. Paired with brine drainage and the resulting decrease in density the sea ice with the new snow-ice layer on top is likely to be lifted out of the water a bit plus, as I wrote above, there might be several millimeters if not centimeters of slush that are above the water line that are refreezing as well (possibly these are the parts that freeze first). Therefore, the resulting sea ice freeboard might very likely not be zero but slightly positive.*

**Reply**: we totally agree with the referee's comment. Indeed the rebalance of the sea ice-snow mass in the case of inundation is pretty complex and the zero ice freeboard is not guaranteed (nor properly defined either). Therefore, the sentence is revised, so as not claiming "zero ice freeboard" after the re-balancing of the sea ice-snow medium.

Additionally, a sentence is added for clarification: "Complex processes occur at the snow-ice interface, such as the further intrusion of sea-water in the snow pack, as well as the gradual drainage of brine within the newly formed snow ice."

*L40: Isn't ice porosity rather related to its air content while here you are talking about the increased permeability due to the widening of brine inclusions and channels?*

**Reply**: yes, porosity is indeed defined as the air content in the ice. And higher porosity is usually associated with better permeability. Temperature (especially near freezing point) has a large role in modulating the porosity. For revision, we have added "permeability" after "porosity", since permeability is directly related to brine movement.

*L42: "brine drainage channels" --> Have such channels been observed in snow on Antarctic sea ice? It would have thought this is a feature of sea ice only.*

**Reply**: the brine drainage still exists in the slush layer, so for revision, we revised it as "brine drainage in the slushy layer".

*L43: What I am so far missing in this paragraph is i) the process of surface melting, percolation of melt water and formation of (fresh) snow-ice at the bottom of the snow pack and ii) rain-on-snow events that can cause any kind of snow property variations - including surface glaze. Most of these you mentioned already in the sentence ending here but you did not connect these well to the processes.*

**Reply**: we reorganize the sentence to differentiate the surface processes and internal processes of the snow cover: "In addition to brine-wetted snow, other factors during winter such as the atmospheric forcings, including precipitation variability, strong winds, and repeated melt/refreeze cycles contribute to the snow's complex stratigraphy. These factors result in variations in snow grain size and density, melt water percolation and refreezing within/under the snow cover, as well as the formation of ice lenses."

*L57: I believe you could (and should) increase the list of sea-ice quantities which retrieval from satellite remote sensing data is influenced by these property changes by i) sea ice motion (e.g. Lavergne and Down, https://doi.org/10.5194/essd-15-5807-2023) and ii) sea ice type (e.g. Melsheimer et al., https://doi.org/10.5194/tc-17-105-2023)*

**Reply**: both sea ice type and motion works are added.

*L99: "above the snow/sea ice interface" --> For the initial installation of the snow buoys this is correct but later on the snow-ice interface migrates towards the ultrasonic sensors due to snow-ice formation. It might make sense to be more specific here in this regard therefore.*

**Reply**: we add "at deployment" to be more accurate.

*L102: "from ... 2017" --> Really? You only used data from two specific days? Or did you want to refer to the period April 30 2016 to January 1 2017?*

**Reply**: revised to "we use data collected during the period from …".

*L127/128: "Vertical .... (2017b)" --> Compared to other publications the degree of detail provided here is relatively low. My questions are:*
*i) were the samples used for the density also used for the salinity measurements or were additional samples taken? If so, how close to the density samples and at which vertical spacing?*

**Reply**: We apologize for omitting detailed information about the dataset. The density and salinity data were both collected from ice stations during PS81 (Paul et al., 2017a, 2017b). Both parameters were measured at 3 cm vertical intervals. However, salinity measurements were less frequent than density measurements. This means that whenever a salinity measurement was taken, a corresponding density measurement was always available at the same depth and location. Initially, we collected all density and salinity measurements to understand the general conditions. We then specifically selected paired density and salinity measurements, taken at the same time and depth, to characterize the snow properties in each snow type. We have revised the text to clarify this process:

"During PS81 (Paul et al., 2017a, 2017b), density and salinity profile were collected at 3 cm intervals. While salinity measurements were less frequent, they were always paired with density measurements at the same depth. We first analyzed all collected data, then specifically examined paired density and salinity measurements to detail snow properties in each stratigraphy."

*ii) How large was the snow volume that was melted? At which temperature did the melting occur? At which ambient temperature where the salinometer measurements carried out?*

**Reply**: According to Lemke et al. (2014), during ANT-XXIX/6, each snow pit included profiles of temperature, salinity, density, and liquid water content, as well as estimations of snow stratigraphic parameters such as snow hardness, grain size, and crystal type for each layer in the snowpack. As mentioned in Frey et al. (2020), these snow pit profiles should be sampled at 2 cm depth resolution using a custom-built, cylindrical, stainless-steel sampling tool with a sample volume of approximately 60 cm$^3$. Frey et al. (2020) also noted that "One set of snow samples was melted on board RV *Polarstern* to measure aqueous conductivity using a conductivity meter (SensIon 5, Hach, salinometer) with a measurement range of 0–200mS cm$^{-1}$ and a maximum resolution of 0.1 µScm$^{-1}$ at low conductivities (0–199.9 µScm$^{-1}$)."

*iii) What type of a salinometer was used and what is its measurement accuracy and sensitivity?*

**Reply**: according to Frey et al., (2020), they used SensIon 5, Hach "with a measurement range of 0–200mS cm$^{-1}$ and a maximum resolution of 0.1 µScm$^{-1}$ at low conductivities (0–199.9 µScm$^{-1}$)."

*iv) As you described yourself, the snow cover can be very heterogeneous. How did you treat visually discernible strong gradients in density such as caused, e.g. by an ice lens buried in the snow?*

**Reply**: As an external user of the ANT-XXIX/6 datasets, we don't know how the staff working on the ice differentiate the snow density from different causes, since the 3 cm interval is an appropriate way to detail the snow stratigraphy even within the strong visualized density gradient. Therefore, the observers did not aim to exclude snow density caused by different factors, but to record and measure the snow density layer by layer. However, according to Arndt et al. (2018), the density cutter may miss hard ice layers within the snowpack. As a result, the mean snow density values might be slightly underestimated.

*v) How did you treat slush at the snow/ice interface if there was any?*

**Reply**: according to the Lemke et al., (2014), "Different types of surfaces (on snow, on ice, on slush) are typically investigated, together with ancillary measurements…". Unfortunately, we cannot access the details in how the staff worked on the ice and how to take the slush samples.

*Weren't there any temperature measurements?*

**Reply**: according to the Lemke et al., (2014), "Every snow pit comprised temperature, salinity, and density profiles as well as liquid water content profiles in addition to the

estimation of snow stratigraphic parameters such as snow hardness, grain size and crystal type for each layer in the snow pack."

*There is also a typo: "now" --> "snow"*

**Reply**: corrected.

*L130-135:*

*What is the motivation to take samples from earlier cruises (2004, 2006) into account?*

**Reply**: we would like to clarify that since comprehensive snow stratigraphy data around the Antarctic are limited, we aim to include all available open-access snow pit data from the Weddell Sea. This approach will allow us to examine if there are any trends in snow stratigraphy over the years. Therefore, we will also include samples from earlier cruises.

*How were the layers defined? Visually or my means of the density observations described further up?*

**Reply**: According to Arndt et al. (2018), the snow grain type and size (e.g., rounded crystals, faceted crystals, depth hoar) were examined with a millimeter-scale grid card to identify the main grain size and type for each layer.

*Layer hardness was recorded using standard measures, I assume?*

**Reply**: As mentioned in Arndt et al. (2018), the hardness of each layer in the snow pits was determined by hand-testing the penetration resistance of the snow. The hardness was categorized into six different classes: very soft (fist; F), soft (3-4 fingers; 4F), medium (1-2 fingers; 1F), hard (pencil; P), very hard (knife blade; K), and ice (I).

*L143/144: "Snow density and salinity ..." --> Were the devices used here of the same type as for the measurements carried out during the Polarstern expeditions? How about the measurement accuracy and sensitivities? How about the means by which snow stratigraphy and grain size were observed. Were these also the same as described above?*

**Reply**: we would like to mention that a similar method was used in Nicolaus et al. (2009), where snow density was determined from volumetric measurements (volume: 500 mL; diameter: 5 cm) in continuous profiles to facilitate vertical averaging. Snow salinity was measured on all 52 volumetric snow density samples by melting them at room temperature and using a salinometer to detect the salinity. We acknowledge that

measuring snow density, wetness, and salinity is challenging, and individual measurements are subject to high uncertainties. Further documentation and analysis are needed to better understand the accuracy and sensitivities of the instruments and measurement methods. Additionally, the presence of ice layers frequently hampered volumetric measurements.

*L147-151: Please review the instrumentation that was actually used during the OIB flights in the Antarctic over sea ice. I am not convinced that LVIS (or other lidar / laser devices) was used. I would think that the main products derived by Kwok et al. (2012) and available over sea ice from these campaigns rely on observations of the ATM and of the snow radar. Giving a bit more information about these instruments would be good - also, for instance, what the flight altitude has been for the Antarctic OIB flights.*

**Reply**: we confirm that the ATM was used for the OIB campaigns we investigate, and LVIS was not available. Details of the instruments are added to the end of the first paragraph introducing OIB data.

*L152-162: I was wondering whether it would make sense to write a bit more about the accuracy and the limitations of these observations here. After all, OIB measurements are not the truth. There is a snow radar resolution based bias for thin snow covers which are under-represented in the measurements and there are substantial difficulties encountered of sea ice with complex surface topography (i.e. ridged ice).*

**Reply**: we thank the referee's suggestion on providing uncertainty/limitation of OIB data. We have added these information to the end of the second paragraph for OIB introduction.

*L168: Where did you get the L3B SMOS TBs from and which processing version has been used for these?*

**Reply**: the L3B SMOS Tb product is available at:
https://icdc.cen.uni-hamburg.de/thredds/catalog/ftpthredds/smos_tb/catalog.html

*L173: Ok, now you have introduced two SMOS TB products but which of these did you actually use?*

**Reply**: we confirm that both the multi-angle mean SMOS Tb L3B (https://icdc.cen.uni-hamburg.de/thredds/catalog/ftpthredds/smos_tb/catalog.html ) and the per-angle, per polarization Tb of RE07 (available through: ftp://ftp.ifremer.fr) are used.

*L184/185: What is the projection used by the ASI SIC product? Please add this information.*

**Reply**: this information is provided in the paragraph: "polar stereographic projection".

*L191/192: "missing snow value due to leads and melting snow"*

*i) It is not clear where the "melting snow" information is coming from. Does the NSIDC product contain a flag? Is this also a 5-day running mean?*

**Reply**: The NSIDC product provides a 5-day running mean and includes a flag indicating snowmelt. Snowmelt is defined by temporal decreases in relative emissivity between 36.5 GHz and 18.7 GHz.

*ii) It appears a bit strange that you use a coarser resolution product (12.5 km) to detect the influence of leads on "missing snow (depth) values" when you are using a much finer resolved product for the SIC. Please consider re-phrasing your wording here.*

**Reply**: We would like to clarify that the ASI and AMSR-E 89 GHz channel-based SIC product has higher resolution due to the finer resolution at 89 GHz (nominally 3 km x 5 km). However, at the lower frequencies of 18.7 GHz and 36.5 GHz, which are used to derive snow depth, the resolution is much coarser. Therefore, the snow depth product has a resolution of 12.5 km. For the revision, we have changed the wording of how we use this snow depth product (last sentence of the paragraph) as follows:

"The snow algorithm depends on the gradient ratio of the vertically polarized Tbs at 18.7 GHz and 36.5 GHz, and is only reliable over seasonal ice and for dry snow conditions (Markus and Cavalieri, 1998). The snow depth and Tbs product from NSIDC, with a resolution of 12.5 km, are used to interpret surface variability in Section 5.1. The snow condition is flagged as 'Snowmelt' when the relative emissivity between 36.5 GHz and 18.7 GHz decreases within 5 days."

*L194: "To intuitively ..." --> I strongly recommend to back up this "intuition" by citing respective published literature. Since I am not sure what you mean by "small-scale ice surface variability" I cannot assist you further here. But it seems important that you inform the reader in more detail about the scale you are refering to here (millimeter, cms, meters?) and why you want to see which feature in the SAR images. Also, why is this intuitive? Wouldn't OIB ATM measurements be much more intuitive as they provide high-resolution information about the actual surface topography?*

**Reply**: we apologize for the improper word of "intuitively" here. What we meant was visual inspection of the surrounding sea ice condition with SAR data. The sentence is revised as: "In order to study the fine-scale sea ice features within the SMOS footprint, we use SAR images that cover the aforementioned in-situ and airborne measurements".

OIB cross-flight path coverage is still insufficient (at about 250m) regarding the SMOS footprint size of >30km.

*In short: It is not entirely clear what the aim of including ALOS PALSAR imagery is.*

**Reply**: the ALOS PALSAR images are used for subjective investigation of the nearly sea ice conditions, at the scale of SMOS footprint.

*Besides this: This is data from the first ALOS, right? Did you consider to use ALOS-2 PALSAR data? They might overlap with many more of the other data sources you are actually using. e.g. SMOS, SMAP, the OIB flights, the buoy observations and so forth.*

**Reply**: we confirm that these images are from ALOS. We do not have directly accessible ALOS-2 images, unfortunately. We are planning to expand the collaboration so as to analyze more L-band SAR data in the Antarctic region.

*L221: "calibrated negative freeboards" --> not clear what is meant by "calibrated" in this context. How were negative freeboards calibrated? And against which source are these calibrated?*

**Reply**: we do not calibrate the ice freeboard, but infer it with ice and snow thickness measurements. The sentence is revised as: "the presence of brine-wetted snow is indicated by the negative ice freeboard, inferred from snow and ice thickness measurements".

*L236-237: What about the ocean heatflux? Is this set to zero?*

**Reply**: The ocean heat flux is set to 8 W/m$^2$ based on Wever et al., (2020).

*You introduced SNOWPACK as allowing to handle several snow AND sea ice layers. You did not comment in your description about the ice layer(s) you used in SNOWPACK. Did you use a one-layer ice slab?*

**Reply**: yes, we used the one-layer for sea ice in SNOWPACK by default.

*L245: Maass et al. applied this model to Arctic sea ice only, didn't they? How compatible is this for the Antarctic?*

**Reply**: the algorithm in Maaß et al (2013) was indeed developed with Arctic in context. But the algorithm is inherently not limited to Arctic sea ice only. Another example is the recent SMOS thin ice product for Antarctic which also directly applies the radiative model developed for Arctic applications (Kaleschke et al., 2024)

*L284: "only valid ... -3degC" --> Where does this threshold come from? Is this also from Geldsetzer et al. 2009?*

**Reply**: this threshold is revealed by the experiment data in Geldsetzer et al. (2009). Above -3 degC, drastic changes in the magnitude and variance of dielectric properties is witnessed. We have noted the limitation in Eqs. 4 and pointed it out here.

*L329: I note that taking a thickness value for sea ice of 2.5 m is a rather high value for typical Antarctic sea ice (see GC1).*

**Reply**: as suggested by referee 2 and 3 and mentioned in GC1, we reanalyzed the sensitivity test, setting the ice thickness (snow depth) from 0.5m (0.2m) until 3m (0.8m). See GC1 for details.

*L332-334: I am a bit surprised about the choice of the air content of the slush - and also about you choice of slush density. Isn't slush basically a high-density mixture of freshwater ice crystals, brine solution, and liquid water with an even higher density than, e.g. first-year ice? Are there really no slush density estimates in the literature from which you could have taken an actual value rather than assuming a density of snow composed of ice layers? This kind of snow potentially contains some air pores but no liquid water or liquid brine as slush does (see GC2).*

**Reply**: As suggested by the referee, we checked the literature and rephrased the sentence as follows: "Based on Jutras et al., (2016), salinity in snow-ice is treated as: $S_{snow-ice}$= 20 g/kg. Following Saloranta et al., (2000), physical properties of snow-ice density is determined as: $\rho_{snow-ice}$= 875 kg/m$^3$, thus snow-ice conductivity ($K_{snow-ice}$) bulk value is 1.045 Wm$^{-1}$K$^{-1}$ based on Eq. 11."

We also added the following description about snow-ice:

"Snow-ice includes more air bubbles and is very distinctive from the coarser columnar crystal structure of congelation ice. It is also much weaker (Saloranta et al., 2000). Therefore, its physical properties differ significantly from those of snow and congelation ice."

*L347/348: "... depth hoar ... began forming due to rain and higher air temperatures." --> I am not convinced that higher air-temperatures contribute to the formation of depth hoar while at the same time snow wetness increases. Isn't depth hoar formation rather related to strong temperature gradients within the ice-snow system / large humidity gradients within the snow, triggered by particularly cold air-temperatures? Please check the available literature. (GC3)*

**Reply**: thanks for the comments, we now revise the sentence into: "Starting from 14-Sep-2016, depth hoar (due to consistent negative temperature gradient) and melt layers (due to rain and higher air temperatures) began forming,.... "

*L386/387: Isn't wind slab not also a form of the snow cover that is simply formed by the action of the wind - without precipitation events, i.e. from wind-induced snow redistribution?*

**Reply**: thanks for the comments, we now revise the sentence into: "..., often found in the uppermost layer of the snow, mainly results from wind transportations, deposition, and wind packing... "

*L393: Aren't brine drainage channels features developing during the formation of sea ice - mainly during columnar sea ice growth? I would then rather change the formulation towards that brine drainage channels are widened.*

**Reply**: revised to: "These layers form intriguingly; they develop when seawater infiltrates the crevices, widening brine drainage channels, which ultimately saturate the underlying snow".

*I was also wondering whether the flooding through such channels is really the main mechanism through which sea water enters the ice-snow interface for Antarctic sea ice. Isn't the role of lateral flooding, i.e. from the floe boundaries, and flooding through cracks in the ice cover the more prominent mechanism, given the fact that there is considerably less columnar sea ice growth in the Antarctic than in the Arctic?*

**Reply**: we totally agree with the referee's comment on the various ways flooding can occur over Antarctic sea ice. The way seawater enters the snow cover differs drastically

from place to place, depending on floe geometry and atmospheric and oceanic forcings. For the sake of completeness, we have added an additional sentence for clarification: "Additionally, seawater can move laterally from cracks and floe edges."

*L394: The capillary wicking of moisture into the basal snow layer mentioned here requires that there is some liquid water already available at the ice-snow interface; it appears hence to be a consequence of whatever flooding process that might have happened before?*

**Reply**: yes, we agree that the presence of water very near the ice-snow interface is indeed a prerequisite for the capillary wicking.

*L451/452: How large is the land influence on the SMAP / SMOS TBs measured in Prydz Bay in comparably close proximity to the coast?*

**Reply**: as pointed out by the referee, the land is indeed very near the site of the buoy's deployment on the landfast ice as the picture showed below. However, what we are dealing with here is not only Tb, but more importantly, the change in Tb and the relationship w/ snow-ice formation. The surrounding conditions (within the SMOS footprint) are very stable for the period of study, consisting of bareland and several grounding icebergs.

[Figure]

Fig.1 ZS buoy location overlaid by Tb on September 1, 2010

*L469-471: Just a comment: These results using the ASPeCt data are kind of surprising because the ASPEeCt observations are just estimates and are certainly much less realiable than the buoy data with respect to the sea ice thickness and snow depth values.*

**Reply**: we agree that the absolute uncertainty of ice and snow thickness by ASPeCt is much larger than the buoy's measurement. However, the spatial representation of ASPeCt is also higher than that of buoys. Whether the improved representation is the cause of higher consistency between the modeled and the observed Tbs, is planned for future study.

*L507-511: These are lines of a more generic, summarizing character that might better be placed into the discussion of even into the conclusions. Try to avoid repeating the same message several times.*

**Reply**: thanks for the suggestion, the last two sentences are removed for the sake of conciseness.

*Instead, what might perhaps be more interesting to learn - in the context of Fig. 7 - is why the agreement between RADIS-L v1.1 modeled TBs and observed TBs is much better for Ice Stations 4 and 6 than for the other three stations? If you have written this somewhere else, then I am sorry for my oversight.*

**Reply**: we would like to highlight that the uncertainty in sea ice concentration from ASI may contribute to Tb overestimation. For instance, the sea ice concentration in stations 2, 4, and 7 is recorded as 100%, while station 3 has a concentration of 96%. More detailed, small-scale datasets, such as SAR images used in Sec. 5.1, need to be further explored. Additionally, the observed ice thickness and snow depth information input in v1.1 are based on mean values rather than distributions or multiple samples. As shown in Fig. 10, using single-point measurements tends to overestimate the mean Tb value.

*L522: "The overestimation of sea ice concentration is directly observed ..."*

**Reply**: revised to "The overestimation of sea ice concentration in the SIC product is evident from ALOS …".

*I suggest to re-phrase this statement, because the ALOS PALSAR image does not provide credible enough information about the sea ice concentration itself. The fact that you see a mode in the backscatter values at lower values is probably an indication of leads but whether these are open water or covered with thin ice remains unknown at this stage. Hence the sea ice concentration in all PALSAR images shown might be pretty close to 100% within the sea ice cover. You might also bear in mind please, that comparably high backscatter values could be caused by wind roughened open water. While you might not find that in the scenes shown, I recommend to be less blunt with statements such as "averaged -13.1 dB, indicating sea ice" in Line 528. There are other*

*issues you could comment on, that are much more clearly visible in the PALSAR images. Basically all these images show a very clear transition between brighter and darker features in the backscatter. I recommend that you check the literature whether the contrast in L-Band backscatter observed here could have to do with i) surface melting, ii) subsurface melting (you might actually see a wet ice/snow interface at L-Band because it penetrates the snow cover), and iii) ice type (especially the images of Oct. 30 show nicely a bright backscatter area kind of hugging the coast of the Antarctic peninsula which could well be the signature of perennial ice, while further east you find seasonal ice).*

**Reply**: we would like to clarify and revise from several aspects:

(1) The bimodal distribution of sigma0 on Oct-30th, and their distribution highlighted in the maps. Both modes correspond to sea ice (different types), not open water within the ice.

(2) Since the sea ice is generally packed in the region of study, the roughening of small open water in the ice pack due to wind is highly unlikely. The sigma0 on these places should be very low (speckle return at nadir, but very low backscatter for SAR which is slant-looking). These places should correspond to the left tail of the sigma0 PDF (Fig. 8d), especially the mode of sigma0 at -21dB on Oct-29th.

(3) The sentence on L528 is revised as: "In contrast, on 29 October 2010, the lower mode at -20.8 dB mainly arises from leads, while the dominant peak around -13.1 dB which constitutes over 90% in the PDF corresponds to sea ice in the region."

(4) The dark regions (small-scale) within the SMOS footprint could indicate open water, leads, or thin/refrozen ice, which cannot be distinguished using the 6.25 km ASI sea ice concentration data. All of these features have a lower Tb compared to congelation ice, meaning they significantly contribute to Tb overestimation.

*L529-535: You might need to re-phrase this paragraph after having worked on the previous one.*

**Reply**: the whole paragraph is revised as: "In summary, the JRA55 atmospheric reanalysis, combined with Tbs from various AMSR-E bands, indicates a significant presence of moisture in the air and potentially surface melt of the snow cover in the eastern section of the OIB flight path. Furthermore, visual inspection of the HH-polarised ALOS images indicates the presence of leads within the ice pack. However, the sea ice concentration product based on AMRS-E cannot directly resolve these leads, and more importantly, reports the SIC at 96.7$\pm$3% within OIB overestimation (differences > 10K) region. This is significantly lower than what the SAR image indicates. The overestimation of SIC causes positive biases in the simulated Tbs compared to SMOS observation. Much lower L-band Tb is usually associated with (refrozen) leads, compared with the typical sea ice cover. The role of the leads are not accounted for due to limited spatial representation by the OIB scans. This result highlights the need for including small-scale ice variability when comparing multi-scale observations of the sea ice."

*L539-541: In view of Fig. 9 I was wondering whether you also tried to separate the influence of theta_a and theta_w? How do TB values change if you only vary theta_w? How do they change it you vary theta_a? How would you be able to explain the changes modeled?*

**Reply**: As mentioned in the manuscript, we treated the frozen slush as a stage of snow-ice layer, whose physical properties differ from those of snow and congelation ice. Snow-ice includes more air bubbles and is very distinctive from the coarser columnar crystal structure of congelation ice, making it much weaker (Saloranta et al., 2000). We followed the approaches outlined in Saloranta et al. (2000) and Jutras et al. (2016) to determine the physical properties of snow-ice. Density of snow-ice is 875 kg/m$^3$, the thermal conductivity of snow-ice is 1.045 Wm$^{-1}$K$^{-1}$, and salinity in snow-ice is 20 g/kg. While the water/liquid content in snow-ice remains relatively stable, the air content can significantly affect the snow-ice Tbs..

[Figure]

*L542-545: I have to admit that I am surprised to see that the overall reduction in TB between dry snow and 80% of the 0.5m thick snow layer consisting of brine wetted snow is only about 6K. Does this make sense in view of the theory? I am also surprised to see that an increase in the fraction of brine wetted snow actually results in a decrease in the modeled TB. Intuitively, since an increase in brine-wetted snow is associated with more liquid water in the snow (the snow is at least moist, right?), I would have expected an increase in the TB values. Can you explain why an increase in the liquid water content in this case results in a decrease in the TBs? Is this the effect of the salinity? If so, why? GC3*

**Reply**: we would like to clarify that the reference scenario is three-layers (dry snow, brine-wetted snow layer, and ice). As mentioned in Line 542, the air temperature is fixed at -30°C. Using a simplified calculation, if we set the sea ice thickness to 2 m, the snow

depth to 0.6 m, the conductivity for dry snow to 0.15 $Wm^{-1}K^{-1}$, for brine-wetted snow to 0.8686$Wm^{-1}K^{-1}$ (computed according to the model and the specific brine-wetted snow density), and for sea ice to 2.14 $Wm^{-1}K^{-1}$, with the sea water temperature at -1.8°C, we get the following results:

- If the snow is completely dry, the bulk snow and ice temperatures will be -18.7°C and -4.6°C, respectively.
- If 80% of the snow is brine-wetted, the bulk snow and ice temperatures will be -20.6°C and -8.8°C, respectively.

Thus, the 6K decrease from a brine-wetted snow layer is quite reasonable.

Consistent with results from C-band (5.3 GHz) summer snow melt over Antarctic sea ice (Willmes et al., 2006), the wetness of brine-wetted snow is not complete and saturated. Metamorphosed snow with increased grain size, as well as the formation of ice layers, leads to decreased emissivity, enhanced volume scattering, and increased backscatter. This causes the Tb to drop.

Additionally, since the air temperature is fixed at -30°C, the sensitivity study is free from atmospheric energy fluxes/forcing on the snow cover. Therefore, the Tb is largely determined by surface snow morphology, similar to the findings of Willmes et al. (2006) at Station POLarstern (ISPOL) in the western Weddell Sea.

*L545-547: "The inclusion ... more akin to ice than to snow ..." --> I get the impression that the TBs change into contrasting directions.  On the one hand, without a slush layer, TBs decrease with an increase in brine-wetted snow layer thickness (and hence net total amount of water in the snow). On the other hand, with a slush layer, a larger water fraction Theta_w results in a TB increase - hence exactly the opposite development. Why?*

**Reply**: We would like to mention that, similar to the previous comment, snow wetness is not the only cause of Tb changes, especially in the context of Antarctic sea ice applications. This is because:

1. Laboratory Experiment by Lohanick (1993): In an environment with temperatures around -7°C, a significant Tb drop of over 100 K was observed due to snowfall, resulting in a saline slush layer forming at the bottom of the snow at 10 GHz. The extra energy from the seawater eventually caused the slush layer to freeze into a highly emissive frazil ice layer, raising the 10 GHz brightness temperature above its bare ice values. Experiments at 85 GHz showed a smaller Tb drop, indicating that at lower frequencies like SMOS (1.4 GHz), a more substantial reduction is expected.
2. Observations Summary from Garrity (1992): The effects of a slush layer on satellite-measured Tbs are illustrated with data from the Weddell Sea in 1989 (Garrity, 1991). Both the western (second-year ice) and eastern (first-year ice)

Weddell Sea were covered by over 50 cm of snow, with slush observed at the snow/ice interface. The attenuation of microwave energy was small enough for microwave emission to come from the entire snow depth and slush layer, resulting in volume scattering from the snow and absorption loss from the slush layer. Garrity (1992) developed a descriptive snow model and mentioned that when slush is present at the snow/ice interface, Tb decreases due to the high free water content (greater than 40%) of the slush. Free water migrates into the lower levels of the snow cover until a slush layer forms at the snow/ice interface, causing a radiometrically homogeneous layer dominated by the low Tb of bulk water (see their Fig. 16-3, 16-6).

*I also note that the increment by which the TBs increase, is substantially larger in case of a slush layer present than without a slush layer present (particularly in case of the slush layer with the highest air content and the lowest water content). How can this be explained? GC3*

**Reply**: We want to clarify that, as mentioned in Line 541, the snow-ice (a stage of snow-ice) layer is determined by the percentage of snow-ice within the brine-wetted snow. Therefore, the more extensive the snow-ice layer, the less brine-wetted snow remains atop. Furthermore, as noted by referee 2 and in Line 304, slush is a transient phase that quickly transforms into newly-formed snow-ice. Consequently, snow-ice acts as a third medium distinct from snow and congelation ice. This distinction is likely the main reason why the inclusion of snow-ice results in larger differences than expected.

*L548/549: "A larger slush depth ..." --> Here you are referring to the non-linear decrease in TBs with increase slush layer depth, right? Do we understand why this decrease is non-linear? What is the physics behind this observation?*

**Reply**: We would like to clarify that when snow-ice forms, the overall thermal conductivity of the snow and sea ice decreases. Under the assumption of quasi-thermal equilibrium, the temperature at the snow-ice interface (now between snow and snow-ice) decreases, moving closer to the surface temperature, which is usually much colder than seawater. This temperature decrease is influenced by the relative depth of the snow-ice layer, the complex permittivity and emissivity of each layer, and the integrated emission at the interfaces. This results in a non-linear relationship between Tb and sea ice thickness/snow depth.

*L568-570: So the TB decrease is about twice as large for an increase in snow salinity than for an increase in snow density. This is interesting. I was wondering, however, why you begin with a snow salinity of 2 g/kg? I was also wondering to which degree the ranges you used are representing typical conditions. Finally, I was wondering, how much the snow salinity is decoupled from the snow moisture. Are we talking about (completely) dry cold snow? Or could it be that a certain (unknown) fraction of the decrease in TBs is*

*due to snow wetness / moisture? One possible way to answer this last question would be to look into results where the snow salinity is zero but the wetness / moisture is not. GC3*

**Reply**: As suggested, the figure below now starts with 0 salinity. The updated Figure 10 includes scenarios with various ice and snow depths. In this analysis, the salinity and wetness of snow are treated independently. In the v1.1 model, we consider both wetness and salinity. Here, we address the scenario where snow salinity is zero but wetness is present, referred to as 'superimposed ice' snow. This type of snow forms during austral spring and summer when the snow cover melts from the top due to warm temperatures, sunlight radiation, or direct rainstorms, and when internal snowmelt water percolates to the colder snow/ice interface where it refreezes (Arndt et al., 2021). In any case, snow stratigraphy during spring and summer is quite complicated and beyond the scope of this paper.

[Figure]

*L575/576: I have a problem with understanding the explanation given here. My understanding of the insulating capabilities of snow so far was that the less dense and the drier the snow is the better it insulates. Hence a very fluffy, 5 cm thick snow cover might insulate better than a 20 cm thick, coarse grained, high density (but still dry) snow cover. However, once the snow cover is wet and/or saline, shouldn't it insulate much less well? Please clarify. GC3*

**Reply**: As mentioned by the referee, compared to new snow (thermal conductivity at 0.07 $Wm^{-1}K^{-1}$), coarse-grained snow, such as hard wind slab and depth hoar, has a larger thermal conductivity (0.35 $Wm^{-1}K^{-1}$), meaning they insulate less effectively than the new snow. Furthermore, as noted in Geldsetzer et al. (2009), the conductivity, and thus the dielectric loss, of brine-wetted snow depends on the concentration of dissolved salts in the brine inclusions and the connectivity between these inclusions.

1. When the brine-wetted snow layer is shallow compared to the snow and ice and at low saturation, the shape of water inclusions becomes needle-like. We found that changes in vertically integrated emissivity are non-monotonic with salinity

and density in the low brine-wetted snow portion due to poorer connectivity between brine inclusions, resulting in lower conductivity in brine-wetted snow.

2. The method we used to calculate the dielectric constant of brine, which varies with temperature, shows a peak at -8°C as modeled by Morey et al. (1984). When the bottom brine-wetted layer is shallow, the temperature of the brine-wetted snow is between -7°C and -9°C in the sensitivity cases, further indicating non-monotonic behavior. Although the depolarization in the dielectric constant is complex and requires further study, this phenomenon also contributes to the observed variability.

To clarify these points, we revised the sentence as follows:

"The likely explanation is that the low conductivity, attributed to the needle-like shape of brine inclusions within the thin brine-wetted layer, disrupts the connectivity of these inclusions (Geldsetzer et al., 2009), resulting in higher temperatures within the snow. Furthermore, the permittivity of that layer becomes highly sensitive to temperature variations around -8°C (Morey et al., 1984), exhibiting larger Tb variabilities in thinner layers."

*L612/613: "basal snow ... ice-surface flooding" --> Please check the causality here. Isn't it the other way round? Flooding of the ice-snow interface leads to basal snow layers having a high salinity and/or moisture content?*

**Reply**: we revise "results in" into "entails" which is more proper since it does not imply causality.

*L616: "Interestingly ... land-fast ice regions" --> Really? In which of the land-fast ice regions in Antarctica did you observe ice-snow interface flooding?*

**Reply**: we would like to point out that the buoy (ZS-2010) used in Sec. 4.2.1 was deployed on landfast ice and it had indeed witnessed snow-ice formation. The main reason is probably due to the mechanical interaction and binding with nearby landfast ice and associated water-level fluctuations based on Li et al., (2023).

*L650-659: Undoubtly it is necessary to point out the limitations of the ASPeCt data set. But I have difficulties to understand why you mention in this context "like the algorithm use, satellite sensor, observation technique" in Line 651. These are all visual, manual ship-borne observations from the ship's bridge that should follow the ASPeCt protocol. The largest uncertainties are the observers themselves and the fact that the ships tend to follow sea ice that is easily navigable, hence avoiding thick, compact and/or ridged sea ice as much as possible, ending up in a negative biases in both hi and hs with respect to the "general" conditions.*

**Reply**: according to the referee's comment, we revise the first sentence of the paragraph to: "The accuracy and statistical parameters of ship observations, such as those from ASPeCt, vary based on factors like the observers' subjective judgements, observation techniques, time of the expedition, and the ships' routes".

*I am also sure that nobody would use "ship-based measurements for validating Tbs" (Lines 657/658), and I also doubt you did this. I understood your usage of ASPeCt observations as a means to provide more observations you can feed into your radiative transfer model, hence you are simply broadening the data basis. I suggest to condense this paragraph according to your specific usage of the ASPeCt data and the specific limitations that results from that and leave it with that.*

**Reply**: according to the referee's comment, we shorten the following part of the paragraph as: "For example, the ships tend to stay easily navigable water, inducing preferential sampling and underestimation of both the sea ice concentration and the thickness (Worby et al. 2008; Weissling et al., 2009). In this study we mainly utilize available data from ASPeCt to broaden the coverage in the vast area of the Southern Oceans. The limitations for using ship-based measurements for model validation need to be examined in detail, especially the effect of uncertainties in the sea ice and the snow thickness parameters".

*L626-634: "In particular ... and retrievals." --> I was wondering whether you could not substantially shorten these lines because it has been well known since the early days of sea ice thickness retrieval using SMOS that small variations in sea ice concentration play a crucial role. Hence you could simply write that the large sensitivity of L-Band Tbs to the presence of open water requires to work with sea-ice concentration data sets of an as fine as possible spatial resolution - such as the one suggested by Ludwig et al (2019) or based on SAR. --> one sentence is enough here, I guess.*

**Reply**: as suggested by the referee, we shorten this part of the paragraph into the following sentences: "In particular, the large sensitivity of L-Band Tbs to the presence of open water requires to work with sea-ice concentration datasets of an as fine as possible spatial resolution - such as SAR based ones as suggested by Ludwig et al (2019)".

*L629: How do "the turbulent flux exchanges" influence the "surface Tb values at L-Band?*

**Reply**: we would like to clarify that: the turbulent flux exchanges influence the surface Tb values at L-band by modulating the surface temperature and moisture content. Sensible and latent heat fluxes transfer heat between the surface and the atmosphere, causing

variations in surface temperature. These temperature changes, in turn, affect the emissivity of the surface, leading to variations in the observed Tb values.

*L641-649: I am not so sure your work points into this direction and I have to admit that these lines are very generic. Of course we need more measurements and they need to be more detailed and we need to do both, field and laboratory studies. But which parameters do we need to observe in a contemporary manner over which spatial and temporal scales with which accuracy to make progress?*

**Reply**: We want to clarify the statement that:

1. Current snow measurements are quite sparse in terms of parameterization and validation for satellite retrieval products.
2. These measurements should not be limited to in-situ/buoy observations but should also incorporate systematic airborne/drone campaigns. These campaigns serve as a crucial bridge between in-situ and satellite measurements, enhancing the applicability of in-situ data and making them more appropriate for satellite validation.

Furthermore, we want to emphasize that these measurement strategies are not universally suitable for all sea ice and snow parameters, as each parameter has its own representative scale and requires specific scaling analysis.

*L645: "most notably in regions with thinner ice" --> this part of the statement is not backed up sufficiently well by your manuscript since the majority of your results are based on modeling using 2.5 m thick ice.*

**Reply**: regarding this comment and previous general comments from the referee, we have now included a more proper set of sea ice/snow thickness parameters that fits the scenarios in the Southern Oceans and the formation of snow-ice. Results are compiled into the revised version of the manuscript, with some shown in this reply.

*Fig. 10 a) and b): I am curious how the curves shown continue towards zero snow salinity and a snow density of, say, 100 kg/m3.*

**Reply**: the revised figure (sub panels) are updated to show a broader range of input parameters for Snow salinity (0 to 10 g/kg, left) and Snow density (100 to 400 kg/m3, right), as shown below.

[Figure]

*I was wondering, whether the snow density shown in Fig. 10b) refers to the snow above the brine wetted layer or to the entire snow layer? If the latter, how realistic is it to assume the same snow density for fresh and saline snow if you are considering brine-WETTED snow? Shouldn't the snow densities be considerably larger for the latter case?*

**Reply**: we hereby confirm with the referee that the snow density in the model is currently independent of the salinity content. Physically they are inter-linked (with certain dependency). But in the current implementation of the model, they are kept independent, and further improvement can be made for better constraining the model. Fig. 10 (a and b) is used for sensitivity study only in the manuscript.

*Fig. 10 c) and d): You seem to have chosen a constant proportional relationship between sea ice thickness hi and snow thickness hs. hs is always 10% of hi. Why? Does this reflect actually encountered conditions? How would the violin for hi=2m look like if you would have used 0.6 m snow thickness? How would violin plots for more realistic Antarctic sea ice thickness values of 0.5 m to 1.5 m look like for the same range of snow thickness values?*

**Reply**: according to this comment (also the GC1 from the referee), we have fully revised Fig. 10 (c and d) to be representative of the Antarctic sea ice in terms of sea ice thickness and snow depth. We reattach the updated panel as follows (also present during the reply to GC1):

[Figure]

*Typos / editoral comments:*

*L20: "underscores" --> "underscore"*

**Reply**: corrected.

*L37: please check: "capillary action" or "capillary suction". I learned it is the latter.*

**Reply**: corrected to "capillary suction".

*L46/47: "affecting ... parameters" --> perhaps better: "affecting the retrieval of various sea ice quantities." ?*

**Reply**: revised to: "affecting its microwave emissivity and the retrieval of various sea ice parameters".

*L53: "Comiso et al., 1997" is focussing on sea ice concentration algorithm intercomparison. While it might mention these processes I was wondering whether there isn't a more specific publication you could cite here in which these processes are detailed from the viewpoint of in-situ observations or radiative transfer modeling results.*

**Reply**: As suggested by the referee, we changed the reference from Comiso et al., (1997) into Ulaby et al., (2014).

*L95: "sea ice temperature" --> perhaps better: "the temperature profile in sea ice and its snow cover" ?*

**Reply**: revised.

*L111+ You might want to include the years during which these buoys operated.*

**Reply**: the years' information (2010-2018) is added, with a reference to Tab.1 where detailed information is provided.

*L113: ",identified" --> ", identified" (blank missing)*

**Reply**: corrected (problem caused by LaTeX).

*L116: You distinguish between an "acoustic sounder to track the distance to the snow surface" and an "underwater sonar altimeter" to track the distance to the ice bottom. Both sensors work with acoustics and both are operated such that one derives a distance. But whether it is warranted to call one "altimeter" I don't know and seems not common to me.*

**Reply**: as suggested by the referee, we remote "altimeter" from the sentence. Indeed it is not usually used for sonar-based sea ice observations.

*L117/118: "temperature string" ... so these buoys indeed do not also use a thermistor string?*

**Reply**: revised to be more precise, as suggested by the referee.

*L138: Since you have described other snow pit measurements already further up, you might consider to begin this paragraph with "Additional" instead of "The"*

**Reply**: revised as suggested.

*L169: You might want to add "at 70 degrees Southern or Northern latitude" since the actual size of the NSIDC grid cells changes with latitude away from the tangential plane used for the projection.*

**Reply**: revised by adding "at 70°N/S".

*L190: ".This" --> ". This" (blank missing)*

**Reply**: corrected (problem caused by LaTeX).

*L252: "over the Arctic" --> perhaps better "in the Arctic" or "for Arctic sea ice"*

**Reply**: revised.

*L305: "brine" --> "saline"*

**Reply**: the whole sentence is revised.

*L316: "determined by" --> "determined following"*

**Reply**: revised.

*L331: "i, a, w are" --> I guess the epsilon is missing here?*

**Reply**: information for epsilon is added.

*L343: "consistent" --> Did you mean "constant"?*

**Reply**: corrected.

*L369-372: "a central tendency defined by a mean value of" --> perhaps better "a mean value of"?*

**Reply**: revised.

*Is the mean value you mention here in fact the "column-averaged value" mentioned in L372? Please clarify.*

**Reply**: The 'column-averaged' value here is the mean density of the snow cover (including all snow types), which is $278.7 kg/m^3$.

*Is the density value of 396.7 kg/m3 the value that has been computed for the rounded crystals / snow-ice / slush cases? This is not entirely clear from your writing.*

**Reply**: based on suggestion, we change the whole sentence to "However, the density of rounded crystals and snow-ice/slush, which also include salinity records, has an average 396.7 kg/m³ but can exceed 600 kg/m³. This makes them significantly denser than the bulk mean values of 280.3 kg/m³ and 309.3 kg/m³, respectively."

*L382: "frequencies" --> I suggest to try to find a different expression here because what you are describing in this subsection is the vertical distribution or variability of the snow stratigraphy.*

**Reply**: based on the suggestion, we change this section title to "Statistics of snow stratigraphy".

*L385: "region(Massom" --> "region (Massom"*

**Reply**: revised.

*L395: "Thus, .." --> Why "thus"? Did you mean "subsequently"?*

**Reply**: revised.

*L397: "ICE" --> Why do you use capital letters here?*

**Reply**: corrected.

*L398: "However ..." --> perhaps better: "In addition, ..."*

**Reply**: revised.

*L411: "declining trend in Tbs" --> I suggest to write either "negative trend in Tbs" or "decline in Tbs" because a "declining trend" suggests that the trend value itself is decreasing.*

**Reply**: changed to "decline in Tbs".

*L415/416: "nearly constant ice concentration approximating 100%" contradicts "increase in open water" --> please check and correct your writing.*

**Reply**: we revise the sentence to be more accurate: "Due to the nearly constant ice concentration approximating 100\%, the Tbs reduction from 243.8 K (11-Sep-2013) to 226.1 K (21-Oct-2013) cannot be attributed to the increase in open water."

*L500: "contrasted" --> Where is the contrast here? If this was the maximum median snow depth then you might write so.*

**Reply**: we would like to explain that by "contrasted", we mean that the relatively thin ice thickness of 1.38m is accompanied by the very thick snow cover (0.48m).

*L502: "more substantial"? --> perhaps better: "larger"*

**Reply**: revised.

*L503: "with and devoid" --> "with and without"*

**Reply**: revised.

*L504/505: "When compared to ... consistently demonstrated an overestimation bias of 8.8K" --> perhaps more simple: "Compared to ... are biased high by 8.8K."*

**Reply**: the whole sentence is revised as: "Compared to SMOS-derived Tbs observations, RADIS-L v1.0 simulations are consistently biased high by 8.8 K. "

*L518: "incorrect sea ice concentration value" --> I suggest that you quantify this better by stating whether the sea ice concentration was too high or too low.*

**Reply**: we have changed "an incorrect sea ice concentration value" to "that the sea ice concentration based on AMSR-E is too low".

*L546: "and lower air content" --> you could add "and hence higher density"*

**Reply**: we revise "higher water content (and lower air content)" into "higher water content, lower air content, and hence higher density".

*L572-574: Just for my clarification: By this percentage you mean a larger fraction of the snow cover that is composed of brine-wetted snow, right? You are not referring to a higher brine volume fraction.*

**Reply**: we confirm that the percentage refers to the vertical range (i.e. depth) in the snow cover that contains brine, NOT the brine's volume in the snow-ice.

*L581-586: Please check these lines. Something seems not to fit well in the context of the "However, Further, ..."*

**Reply**: we have corrected the sentence on l583.

*In this context: Did you think about that the thinner the brine-wetted snow layer is, the higher is the likelihood to receive a signal from the sea ice underneath?*

**Reply**: we fully agree with the referee that if the brine-wetted layer is too thin, it will appear less opaque at L-band, revealing the signal below (i.e., sea ice). This effect is indeed captured by the model (RADIS-L, v1.1).

*L598: "ice thickness deepens" --> I guess a snow layer can deepen (even though I like to talk about snow thickness as it is simply the vertical extent between the snow surface and the underside of the snow and hence similar in definition to that of the sea ice thickness) but not an "ice thickness". So maybe rather write "increase"*

**Reply**: we agree with the referee, and changed "deepens" to "increases".

*L602: Why "dramatic"? While we know a lot about the Arctic sea ice thickness from submarine and moored sensors in addition to the satellite observations a thickness decrease is present, yes, but I would not call it "dramatic" - especially during the past 5-10 years when, for instance, the PIOMAS time series of the Arctic sea ice volume shows a plateau of stagnating values rather than a continuation of a decrease. And for the Antarctic, we know much less about the past sea ice thickness distribution and perhaps should not come up with adjectives like "dramatic". What we do know is that the Antarctic sea ice cover is substantially more variable than the Arctic one.*

**Reply**: according to the referee's suggestion, we replace the improper adjective of "dramatic" to "drastic".

*L614: One ")" can be deleted.*

**Reply**: the extra ")" is removed.

*L618: By "changes in the depth" I would understand changes of at which point, when measured from the snow surface, the region of brine-wetted snow begins. But what I guess you want to say here is "changes in the thickness and/or vertical extent"*

**Reply**: we have revised it according to the referee's comment: "changes in the thickness and/or vertical extent".

*L623: By "reanalysis-driven" you refer to atmospheric re-analyses? Not entirely clear.*

**Reply**: yes, we meant "atmospheric reanalysis". It is revised accordingly.

*L625: "radiative properties of ice surfaces." --> Suggest to add: "at L-Band frequencies."*

**Reply**: added as suggested by the referee.

*L626: You are referring to the modeled Tb values here? Then I suggest to add "modeled".*

**Reply**: added "modeled".

*"surface Tbs values" --> "surface Tb values"*

**Reply**: corrected.

*L637: "properties" --> please mention which properties you refer to here.*

**Reply**: revised to: "the properties of the slush layer such as thickness and brine volume"

*"depth" --> "thickness"*

**Reply**: revised.

*Fig. 3 caption: What is a "white dost"?*

**Reply**: it should be "dots". Revised.

*Why do you write "ICE crust" instead of "Ice crust" at the x-axis annotation of panels a/b) and d)?*

**Reply**: the label of Fig. 3d is corrected.

*"Decomposing and Fragmented" in d) is missing in a) and b)? Also, did you mean "Decomposed"?*

**Reply**: The snow density and salinity were collected from Paul et al., (2017 a and b), where no decomposed and Fragmented were identified from them. And we revise "Decomposing" to "Decomposed" in the figure.

*Fig. 6: If panels b) and c) are heat maps (please correct the caption) then you need to provide a legend which translates to color into counts.*

**Reply**: the corresponding colormaps are added to panel b and c of Fig. 6.

*Fig. 6 caption: I suggest to write "ASPeCt observations" instead of "ASPeCt measurements".*

**Reply**: revised according to the referee's suggestion.

*Fig. 7 left y-axis: "freeboard thickness" does not make sense. Please correct accordingly into "Sea ice/snow thickness & sea ice freeboard" as this is what you want to refer to.*

**Reply**: the label of the y-axis is revised accordingly.

*Also the first line of the caption needs "thickness" to be added behind "sea ice" and "snow" plus "sea ice" to be added in front of "freeboard".*

**Reply**: revised according to the referee's suggestion.

*Fig. 8: It might be a matter of taste but I would prefer to have the images of the earlier date to the left (Oct. 29) and those of the later date to the right (Oct. 30).*

**Reply**: the layout of the panels of Fig. 8 is revised. Now the images of the earlier day (Oct-29) are on the top, while those on the later day (Oct-30) are on the bottom.

*Fig. 9: The legend within the figure lacks the line for 80% percentage of brine-wetted layer.*

**Reply**: the line for 80% brine-wetted layer in the legend is now added.

*I suggest to make clear in the caption that the overall snow depth used is 0.5 m and that the "slush layer depth" is a "relative slush layer depth" or perhaps even better a "relative slush layer thickness".*

**Reply**: we fully agree with the referee's comment which will make it much clearer. Based on the above revision, an additional sentence is added: "The total depth of the snow cover is 60 cm for all cases, while the depth of the snow-ice layer is relative to that of the brine-wetted snow layer."

*Fig. 10 caption:*

*(a) and (c) is salinity and (b) and (d) is density - contrary to what you write in the caption. There are no "e" and "f"*

**Reply**: the caption of Fig. 10 is corrected.

*Fig. B.3: The y-axis is slightly misleading. This is not the "snow height".*

**Reply**: the label of the y-axes is changes to "sea ice/snow depth".

*Fig. B.5: How many data points are shown here? Would it make sense to turn this into a heat map?*

**Reply**: in total there are 866 points. And as suggested by the referee, the figure is changed into a heat map.

*Figure B.6: Which AMSR-E snow depth product is shown here?*

**Reply**: As mentioned in Sec. 2.4.1, snow depth product is based on Markus and Cavalieri, (1998), and is accessible via: https://nsidc.org/data/ae_si12/versions/3

*In the caption you write "SMOS Tbs" but actually you seem to show both SMOS and AMSR-E Tbs; hence in the first line of the caption it needs to read "...between simulated and observed Tbs".*

**Reply**: we would like to clarify that the difference of L-band Tb between simulation and SMOS observation is overlaid against SMOS and AMSR-E Tbs. We have revised the caption to be more clear: "The Tb differences (circles, colored red to blue) between RADIS-L simulation and SMOS observation for the 28-Oct-2010 track, overlaid with: (a) SMOS (1.4 GHz) and different AMSR-E frequencies from (18 ~ 89 GHz), (b) to (e)."

*Fig. B.7: What are the tracks in blue-white-red denoting?*

**Reply**: the colors denote the difference between the modeled L-band Tb and the SMOS observation. The caption is revised to be clearer (see above).

*Since you write the units of the parameters shown below the columns of panels you might not need to repeat the units in the caption. How can a "net precipitation" be negative? Is this perhaps E-P? I note that the scale of the legend of this quantity is not well chosen because large regions are shown with a saturated color. You might consider to change this.*

**Reply**: we confirm that it is P-E. The figure label is revised from "Net precipitation" to "P-E". The colormap is also slightly tuned to be less saturated, as suggested by the referee.

*L907/908: Reference needs to be Studinger, N. K., ....*

**Reply**: the names of this reference are corrected.

**References**:

Arndt, S. and Paul, S., 2018. Variability of winter snow properties on different spatial scales in the Weddell Sea. Journal of Geophysical Research: Oceans, 123(12), pp.8862-8876.

Arndt, S., Haas, C., Meyer, H., Peeken, I. and Krumpen, T., 2021. Recent observations of superimposed ice and snow ice on sea ice in the northwestern Weddell Sea. The Cryosphere, 15(9), pp.4165-4178.

Frey, M.M., Norris, S.J., Brooks, I.M., Anderson, P.S., Nishimura, K., Yang, X., Jones, A.E., Nerentorp Mastromonaco, M.G., Jones, D.H. and Wolff, E.W., 2020. First direct observation of sea salt aerosol production from blowing snow above sea ice. Atmospheric Chemistry and Physics, 20(4), pp.2549-2578.

Garrity, C., 1992. Characterization of snow on floating ice and case studies of brightness temperature changes during the onset of melt. Geophysical monograph series, 68, pp.313-328

Garrity, C., 1992. Passive microwave remote sensing of snow covered floating ice during spring conditions in the Arctic and Antarctic. York University.

Geldsetzer, T., Langlois, A. and Yackel, J., 2009. Dielectric properties of brine-wetted snow on first-year sea ice. Cold Regions Science and Technology, 58(1-2), pp.47-56.

Jutras, M., Vancoppenolle, M., Lourenço, A., Vivier, F., Carnat, G., Madec, G., Rousset, C. and Tison, J.L., 2016. Thermodynamics of slush and snow–ice formation in the Antarctic sea-ice zone. Deep Sea Research Part II: Topical Studies in Oceanography, 131, pp.75-83.

Kaleschke, L., Tian-Kunze, X., Hendricks, S. and Ricker, R., 2024. SMOS-derived Antarctic thin sea-ice thickness: data description and validation in the Weddell Sea. Earth System Science Data Discussions, 2023, pp.1-30.

Lemke, P., 2014. The expedition of the research vessel" Polarstern" to the Antarctic in 2013 (ANT-XXIX/6). Berichte zur Polar-und Meeresforschung= Reports on polar and marine research, 679.

Li, N., Lei, R., Heil, P., Cheng, B., Ding, M., Tian, Z., and Li, B.: Seasonal and interannual variability of the landfast ice mass balance between 2009 and 2018 in Prydz Bay, East Antarctica, The Cryosphere, 17, 917–937, https://doi.org/10.5194/tc-17-917-2023, 2023.

Lohanick, A.W., 1993. Microwave brightness temperatures of laboratory-grown undeformed first-year ice with an evolving snow cover. Journal of Geophysical Research: Oceans, 98(C3), pp.4667-4674.

Ludwig, V., Spreen, G., Haas, C., Istomina, L., Kauker, F. and Murashkin, D., 2019. The 2018 North Greenland polynya observed by a newly introduced merged optical and passive microwave sea-ice concentration dataset. The Cryosphere, 13(7), pp.2051-2073.

Maaß, N., 2013. Remote sensing of sea ice thickness using SMOS data (Doctoral dissertation, University of Hamburg Hamburg).

Markus, T. and Cavalieri, D.J., 1998. Snow depth distribution over sea ice in the Southern Ocean from satellite passive microwave data. Antarctic sea ice: physical processes, interactions and variability, 74, pp.19-39.

Massom, R.A., Eicken, H., Hass, C., Jeffries, M.O., Drinkwater, M.R., Sturm, M., Worby, A.P., Wu, X., Lytle, V.I., Ushio, S. and Morris, K., 2001. Snow on Antarctic sea ice. Reviews of Geophysics, 39(3), pp.413-445.

Massom, R.A., Lytle, V.I., Worby, A.P. and Allison, I., 1998. Winter snow cover variability on East Antarctic sea ice. Journal of Geophysical Research: Oceans, 103(C11), pp.24837-24855.

Morey, R.M., Kovacs, A. and Cox, G.F., 1984. Electromagnetic properties of sea ice. Cold Regions Science and Technology, 9(1), pp.53-75.

Nicolaus, M., Haas, C. and Willmes, S., 2009. Evolution of first‑year and second‑year snow properties on sea ice in the Weddell Sea during spring‑summer transition. Journal of Geophysical Research: Atmospheres, 114(D17).

Paul, S., Arndt, S., and Stoll, N.: Snow density measurements at ice stations during POLARSTERN cruise PS81 (ANT-XXIX/6, AWECS), https://doi.org/10.1594/PANGAEA.881717, [Date Accessed:28-Feb-2022], 2017a.

Paul, S., Arndt, S., and Stoll, N.: Snow salinity measurements at ice stations during POLARSTERN cruise PS81 (ANT-XXIX/6, AWECS), https://doi.org/10.1594/PANGAEA.881714, [Date Accessed:28-Feb-2022], 2017b

Saloranta, T.M., 2000. Modeling the evolution of snow, snow ice and ice in the Baltic Sea. Tellus A: Dynamic Meteorology and Oceanography, 52(1), pp.93-108.

Ulaby, F. and Long, D.: Microwave radar and radiometric remote sensing, Michigan:University of Michigan Press, 2014.

Weissling, B., Ackley, S., Wagner, P. and Xie, H., 2009. EISCAM—Digital image acquisition and processing for sea ice parameters from ships. Cold Regions Science and Technology, 57(1), pp.49-60.

Wever, N., Rossmann, L., Maaß, N., Leonard, K.C., Kaleschke, L., Nicolaus, M. and Lehning, M., 2020. Version 1 of a sea ice module for the physics-based, detailed, multi-layer SNOWPACK model. Geoscientific Model Development, 13(1), pp.99-119.

Worby, A.P., Geiger, C.A., Paget, M.J., Van Woert, M.L., Ackley, S.F. and DeLiberty, T.L., 2008. Thickness distribution of Antarctic sea ice. Journal of Geophysical Research: Oceans, 113(C5).

---

## Author Response (AR2)

The authors would like to thank the editor and the two anonymous referees for their comments on our revised manuscript. Following the comment in **Report #1** from **anonymous referee #3**, we make the following replies and corresponding revisions to the manuscript. The original comment from the referee is in *blue italic*, followed by our reply. Moreover, in the marked version of the revised manuscript, the revisions are highlighted with 'REV'.

**Report #1:**

*i am fine with the overall response and revision. i do have a few more questions for clarifications. (1) as indicated by other referees and you agreed that the ASPeCT should not be used for validation, but you used them anyway and as i mentioned that the ASPeCT data actually show better results than the two other datasets. therefore, should you still keep ASPeCT data as the validation? and how good your model actually modeling the real situation? i would say "it is not that good". (2) for my comment on the L204: what i meant was why you did not correct it using the actual incidence angle for each pixel, while using the constant angle of 35 degree for all pixels. (3) for the explanation of L402-406, i am ok with that. if not already done so in the revision, i hope you can add the detail such as "we compute the daily mean location of the buoy and find the cell of the Tb product that contains the buoy's (mean) location, also use the daily mean buoy data"*

**Reply**: we make the following replies and revisions to the 3 points raised by the referee.

Regarding first point on the use of ASPeCT data for the "validation" of the RADIS-L model, we have revised the title of Sec. 4.2.2, which contains related content, from "***Tbs validation in ship-based and airborne observations***" to "***Inter-comparison of Tbs based on airborne and ship-based observations***". By doing this, we refrain from calling the ASPeCT-based study as validation, due to its potential representation issues.

Regarding the second point of the comment regarding the incidence angle of the SAR image: we confirm that the incidence angle correction is based on the actual incidence angle for each pixel, and all the backscatter values are corrected to the incidence angle of 35-deg. We didn't use (or assume) the constant angle of 35-deg for the original image.

Regarding the third point, we revise the paragraph to contain a more detailed introduction of the methodology, as follows: "***We adopt the following protocol to match buoy's measurements to satellite Tbs. For each buoy, we compute its daily mean locations. The daily Tb map for each daily mean location is used to attain the Tb value in the cell that contains the specific location. Then the buoy's daily measurements are matched to Tbs for further comparison.***"